# Genetic screen in Drosophila muscle identifies autophagy-mediated T-tubule remodeling and a Rab2 role in autophagy

Naonobu Fujita[1,2]*, Wilson Huang[1], Tzu-han Lin[1], Jean-Francois Groulx[1], Steve Jean[1], Jen Nguyen[1], Yoshihiko Kuchitsu[2], Ikuko Koyama-Honda[3], Noboru Mizushima[3], Mitsunori Fukuda[2], Amy A Kiger[1]*

[1]Section of Cell and Developmental Biology, Division of Biological Sciences, University of California, San Diego, La Jolla, United States; [2]Laboratory of Membrane Trafficking Mechanisms, Department of Developmental Biology and Neurosciences, Graduate School of Life Sciences, Tohoku University, Sendai, Japan; [3]Department of Biochemistry and Molecular Biology, Graduate School and Faculty of Medicine, The University of Tokyo, Tokyo, Japan

**Abstract** Transverse (T)-tubules make-up a specialized network of tubulated muscle cell membranes involved in excitation-contraction coupling for power of contraction. Little is known about how T-tubules maintain highly organized structures and contacts throughout the contractile system despite the ongoing muscle remodeling that occurs with muscle atrophy, damage and aging. We uncovered an essential role for autophagy in T-tubule remodeling with genetic screens of a developmentally regulated remodeling program in Drosophila abdominal muscles. Here, we show that autophagy is both upregulated with and required for progression through T-tubule disassembly stages. Along with known mediators of autophagosome-lysosome fusion, our screens uncovered an unexpected shared role for Rab2 with a broadly conserved function in autophagic clearance. Rab2 localizes to autophagosomes and binds to HOPS complex members, suggesting a direct role in autophagosome tethering/fusion. Together, the high membrane flux with muscle remodeling permits unprecedented analysis both of T-tubule dynamics and fundamental trafficking mechanisms.

*For correspondence: naonobu. fujita.b8@tohoku.ac.jp (NF); akiger@ucsd.edu (AAK)

## Introduction

Differentiated muscle cells, or myofibers, are highly organized in order to coordinate the roles of specialized subcellular structures involved in contraction. Myofibril bundles of sarcomeres provide the contractile force. The power of contraction, however, requires synchronous sarcomere function under control of the 'excitation-contraction coupling' system that includes two membranous organelles, the sarcoplasmic reticulum (SR) and Transverse (T)-tubules (*Al-Qusairi and Laporte, 2011*). The T-tubule membrane network is continuous with the muscle cell plasma membrane, with tubulated membranes that invaginate radially inward in a repeated pattern at each sarcomere. With excitation-contraction coupling, neuromuscular action potentials are transmitted along the muscle T-tubule membrane to the SR junction, or dyad/triad, triggering coordinated SR $Ca^{2+}$ release and synchronous sarcomere contractions (*Al-Qusairi and Laporte, 2011*). Formation of organized T-tubule membranes is thus critical for muscle function (*Takeshima et al., 2015*). Mechanisms must also remodel the T-tubule membrane network with ongoing myofiber reorganization in response to muscle use, damage, atrophy and aging. However, the extent and mechanisms of T-tubule

remodeling remain largely unknown, in part due to challenges with observing T-tubule membrane network dynamics within intact mammalian myofibers.

The T-tubule network includes both transversal and longitudinal tubular membrane elements that form and mature with myofiber differentiation and growth. In mouse skeletal muscle, mostly longitudinal tubular membranes initially present in embryonic muscle are remodeled postnatally with expansion to predominantly transversal tubular elements (*Takekura et al., 2001*). In contrast, both longitudinal and transversal T-tubule elements are maintained in adult mammalian cardiac muscle (*Brette and Orchard, 2003*) and in insect muscles (*Razzaq et al., 2001*). Relatively few molecular factors are known to shape the T-tubule network, and perhaps not surprisingly, all of which so far encode for membrane-associated functions (CAV3, DYSF, BIN1/Amph2, MTM1, DNM2) (*Butler et al., 1997*; *Hnia et al., 2012*; *Lek et al., 2012*; *Morlot and Roux, 2013*; *Tang et al., 1996*). Mutations in each also are associated with human myopathy and/or cardiomyopathy with T-tubule disorganization (*Bashir et al., 1998*; *Betz et al., 2001*; *Bitoun et al., 2005*; *Laporte et al., 1996*; *Liu et al., 1998*; *Minetti et al., 1998*; *Nicot et al., 2007*), pointing to the critical importance of membrane-mediated mechanisms to maintain the T-tubule membrane network.

Drosophila is a powerful system for insights into the functional requirements for T-tubule formation and remodeling. The BIN1 BAR-domain protein has a conserved function involved in membrane tubulation required for T-tubule formation that was first described for the single Drosophila homolog, Amphiphysin (*Lee et al., 2002b*; *Razzaq et al., 2001*). The *amph* null mutant flies lack transversal T-tubule element membranes in myofibers at all developmental stages, corresponding with both larval and adult mobility defects (*Razzaq et al., 2001*). In contrast, the *myotubularin* (*mtm*) fly homolog of mammalian MTM1/MTMR2/MTMR1 subfamily of phosphatidylinositol 3-phosphate phosphatases is required only at later stages in development for T-tubule remodeling. While *mtm* loss of function has no obvious effects on larval muscle T-tubule organization or function, *mtm*-depleted post-larval stage muscles lack transversal T-tubule membranes with adult mobility defects in eclosion and flight (*Ribeiro et al., 2011*). Together, the *amph* and *mtm* mutant conditions that both lack transversal T-tubule elements in post-larval stage muscle yet different early development requirements underscores that distinct mechanisms are involved in T-tubule formation (*amph*-dependent) versus maintenance/remodeling (*amph*- and *mtm*-dependent).

In Drosophila, a set of larval body wall muscles that persist as viable pupal abdominal muscles, called dorsal internal oblique muscles (IOMs), are essential for adult eclosion (*Kimura and Truman, 1990*). During metamorphosis, changes in IOM cell size and myofibril content have been noted (*Kuleesha et al., 2014*, *2016*). We previously showed that wildtype IOMs undergo dramatic cortical and membrane remodeling with costamere integrin adhesion complex disassembly and reassembly at discrete pupal stages (*Ribeiro et al., 2011*). In contrast, the *mtm*-depleted IOMs exhibited persistent disassembly or a block in reassembly of integrin costameres along with the loss of transversal T-tubule membranes at late pupal stages, but without any precocious cell death (*Ribeiro et al., 2011*). A striking feature in the *mtm*-depleted IOMs was the accumulation of endosomal-like membranes decorated with integrin and T-tubule markers, Amph and Discs large (Dlg1, a PDZ protein). Altogether, these results suggest that T-tubule membranes may undergo disassembly-reassembly with normal myofiber remodeling, including the delivery of disassembled T-tubule membrane into an endomembrane trafficking pathway. The role for a molecular-cellular program in control of T-tubule remodeling that is at least partially distinct from that involved in initial T-tubule formation raises many questions about possible mechanisms, including the regulation of T-tubule organization and dynamics, the membrane fate(s) and source(s) with disassembly-reassembly, respectively, and the specific membrane trafficking routes and effectors involved. Possible hints may come from studies of other specialized dynamic cell membrane invaginations shown to involve endosomal and Golgi membrane trafficking pathways, such as cellularization of Drosophila syncytial embryos (*Lee and Harris, 2013*, *2014*; *Pelissier et al., 2003*) and the tubulated demarcation membrane system in megakaryocyte platelet formation (*Eckly et al., 2014*).

Membrane trafficking relies on the large family of Rab GTPases, with over sixty Rabs in humans and thirty in flies (*Klöpper et al., 2012*). The different Rabs are under distinct spatiotemporal regulation for recruitment, activation and functions at specific membrane compartments or domains. Guanine nucleotide exchange factors (GEFs) convert specific inactive GDP-bound Rabs to an active GTP-bound form. Active Rab-GTP then recruits a range of specific effector proteins to the membrane that mediate key trafficking functions, including cargo selection, vesicle budding, transport,

tethering and fusion. Subsequently, GTPase-activating proteins (GAPs) deactivate Rabs by promoting GTP hydrolysis. Many membrane compartments have been defined by well-established localized functions of specific Rabs, for example: ER (Rab1), Golgi (Rab1, Rab6), secretory vesicles (Rab8), early endosomes (Rab5, Rab21), recycling endosomes (Rab11, Rab35), late endosomes (Rab7, Rab9), lysosomes (Rab7) and others (*Jean and Kiger, 2012*; *Stenmark, 2009*). Thus, identifying the specific Rabs required for a cellular process can provide clues to potential underlying membrane trafficking mechanisms involved. However, examples exist of Rabs with multiple known sites of function or yet unknown functions, and conversely, certain cellular processes – like T-tubule remodeling – lack defined roles yet for any Rabs.

Here, we utilized the advantages of Drosophila IOMs to screen for Rab GTPases and related membrane trafficking functions required for T-tubule remodeling in intact muscle. Our results show that the entire contractile and excitation-contraction coupling system, including T-tubules, are disassembled and reassembled in IOMs during Drosophila metamorphosis. We found that autophagy, the membrane trafficking process for degradation of cytoplasmic contents by delivery to lysosomes, is upregulated with IOM remodeling where it plays an indispensable role for progression through T-tubule disassembly to reassembly. Our genetic analysis of IOM remodeling also reveals an unexpected and broad role for Rab2 in autophagy in flies and mammals. From our data, we propose that Rab2 localizes to autophagosomes where it interacts with the HOPS complex, which in turn, mediates tethering and trans-SNARE complex formation with Rab7-marked lysosomes to promote autophagosome-lysosome fusion. Together, these results show that Drosophila IOM remodeling provides an unprecedented in vivo context for discovery and analysis of T-tubule dynamics with relevance to human myopathy, as well as an ideal system due to high membrane flux to study fundamental trafficking pathways.

## Results

### Differentiated myofiber remodeling includes regulated T-tubule membrane disassembly and reassembly

To monitor T-tubule remodeling in Drosophila abdominal muscles, we expressed the mCD8:GFP transmembrane fusion protein as a marker of the muscle cell and T-tubule membranes (*Peterson and Krasnow, 2015*). When observed in live IOM persistent larval muscles through the cuticle at 4 days after puparium formation (4d APF), mCD8:GFP showed a mesh-like pattern (*Figure 1A*) that in fixed samples colocalized with Dlg1-marked T-tubules (*Figure 1B–D* and *Figure 1—figure supplement 1A*) and sarcomere Z-lines (*Figure 1—figure supplement 1B*). The brightness of mCD8:GFP enabled us to monitor membrane dynamics during IOM remodeling in undissected live animals. As previously reported (*Kuleesha et al., 2014*; *Wasser et al., 2007*), the IOMs remodeled during metamorphosis with myofiber thinning through 2d APF followed by rethickening from 3-4d APF (*Figure 1E–F*, top and middle rows). Along with these cell morphology changes, the well-organized mCD8:GFP-marked membranes detected in third instar larval precursor muscles were disassembled in IOMs by 1-2d APF and then reassembled by 4d APF (*Figure 1F*, middle and bottom rows).

To more specifically investigate myofiber remodeling, we monitored Dlg1:GFP (T-tubules), Reticulon:GFP (Sarcoplasmic Reticulum; SR), and GFP:actin (myofibrils) at 24h intervals during metamorphosis (*Figure 2A*). Each of these organelles was disassembled by 1-2d APF, and then reassembled by 4d APF in IOMs (*Figure 2A*). Furthermore, ultrastructual analysis of myofiber remodeling by transmission electron microscopy (TEM) imaging of IOM transverse sections (*Figure 2B*) confirmed both the timing and extent of the disassembly and reassembly stages during metamorphosis (*Figure 2C–G*). This reveals that differentiated myofiber structures critical for muscle function, including T-tubule membranes, undergo regulated and stereotypical remodeling in IOMs during metamorphosis.

### A unique T-tubule remodeling phenotype with knockdown of a set of genes

In order to identify functions involved in T-tubule membrane remodeling, we performed a muscle-targeted RNAi screen of candidate membrane trafficking-related genes, including all fly genes

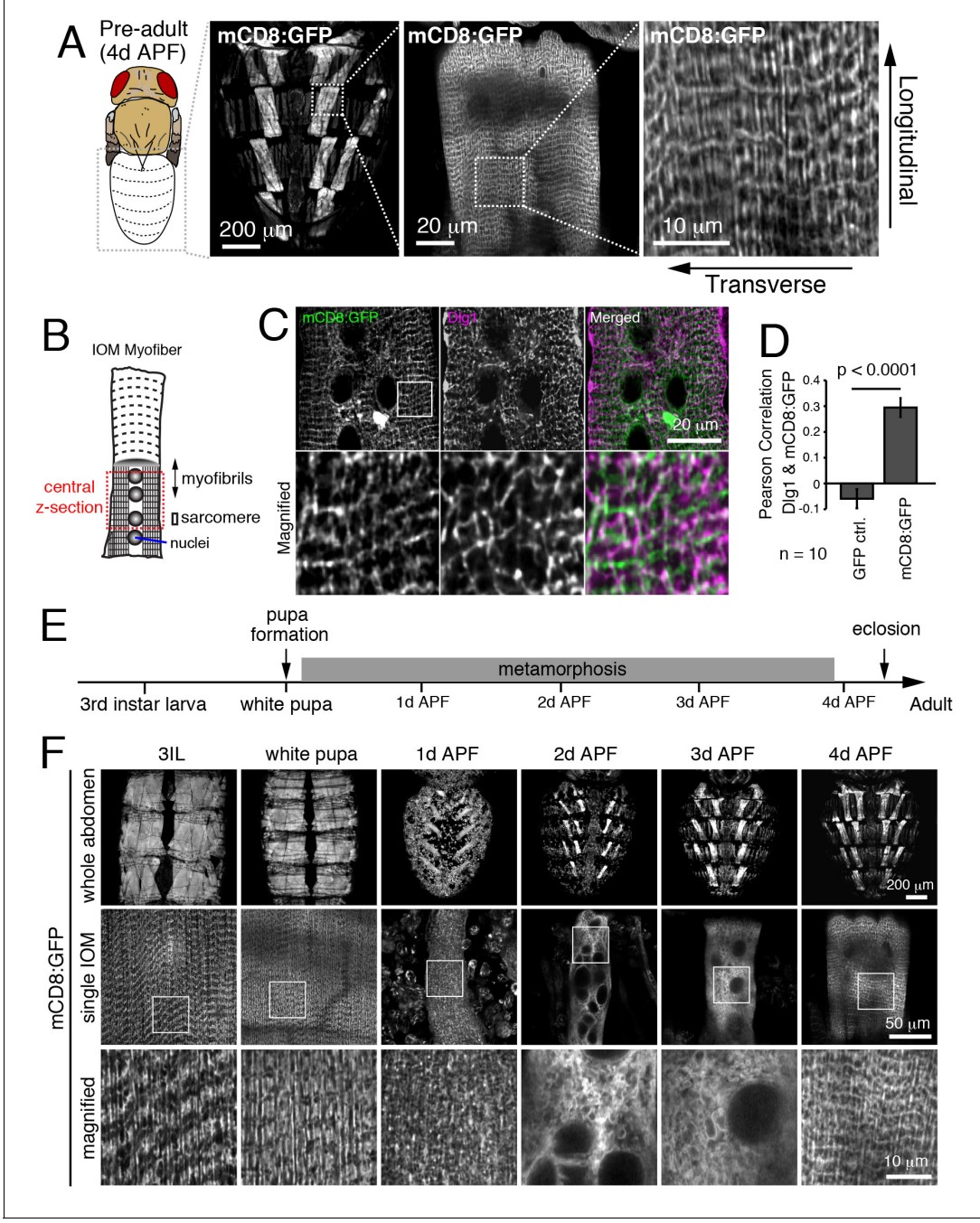

**Figure 1.** Detection of T-tubule membrane organization and remodeling in intact Internal Oblique Muscle (IOM) of live Drosophila. (A) mCD8:GFP showed a mesh-like pattern in pharate/pre-adult dorsal abdominal IOMs at 4d APF by live imaging, with both transversal and longitudinal membrane elements as indicated. (B) Schematic of IOM and z-section regions imaged in panel C. (C–D) Colocalization between mCD8:GFP (green) and Dlg1 (pink) at T-tubules in 4d APF IOMs quantified as Pearson's correlation between Dlg1 and GFP or mCD8:GFP; ± SEM of pooled data for 10 images from three experiments. (E) Time line of fly development from third instar larva to adult at 25°C; days after puparium formation (d APF). (F) Time course microscopy of mCD8:GFP in dorsal muscles imaged through the cuticle of live wildtype animals from third instar larva (3IL) to 4d APF, showing membrane remodeling in abdomens (top), central sections of individual IOMs (middle) and magnified view of boxed regions (bottom). See *Figure 1—figure supplement 1* for related data.

The following source data and figure supplement are available for figure 1:

**Source data 1.** Relates to *Figure 1D*.

*Figure 1 continued on next page*

*Figure 1 continued*

**Figure supplement 1.** mCD8:GFP partially colocalizes with Zormin, a Z-line marker protein.

predicted to encode Rab GTPases, Arf GTPases, sorting nexins, BAR domain proteins, SNARE proteins, and phosphoinositide regulators (*Supplementary file 1*). Since T-tubule organization is required for muscle function (*Al-Qusairi and Laporte, 2011*; *Razzaq et al., 2001*), we first screened for muscle-targeted RNAi effects on fly mobility. Among 300 RNAi lines tested, 77 lines showed a defect in adult eclosion or mobility, 151 lines resulted in normal viability and mobility, and 83 lines were unscored due to pre-adult lethality (*Figure 3A*). As a secondary screen, we tested the 77 RNAi lines with eclosion or mobility defects for mCD8:GFP organization by live cell imaging in IOMs at 4d APF. We identified 10 RNAi lines targeting a set of 5 different genes, representing two Rab GTPases and three soluble N-ethylmaleimide-sensitive factor attachment protein receptor (SNARE) proteins, with a similar phenotype (*Figure 3A–B* and *Supplementary file 1*). Instead of the organized T-tubule network seen in control IOMs, RNAi of *Rab2*, *Rab7*, *Stx17*, *SNAP29* or *Vamp7/8* each resulted in an accumulation of mCD8:GFP-positive small vesicles that filled often misshapen or swollen myofibers (*Figure 3B*). Unlike in controls, T-tubules (Dlg1) and most myofibrils (F-actin) were absent throughout these RNAi-treated IOMs at 4d APF (*Figure 3C–D*).

The 'Rab2 class' of shared phenotypes suggested that these five genes all function in a shared process or pathway in IOMs. Indeed, it has been shown that four of the five genes play known roles together in lysosome fusion: Stx17, SNAP29, and Vamp7/8 form a trans-SNARE complex involved in autophagosome-lysosome fusion (*Itakura et al., 2012*; *Takáts et al., 2013*), while Rab7 functions in late endosome-lysosome fusion as well as late steps in autophagy (*Gutierrez et al., 2004*; *Jäger et al., 2004*; *Hegedűs et al., 2016*). It was unexpected, however, to find a shared RNAi phenotype between this set of known functions and Rab2, which had been implicated with functions at the ER and Golgi (*Saraste, 2016*). The specificity of the Rab2 RNAi phenotype was confirmed by rescue with co-expression of a wildtype Rab2 transgene (*Figure 3—figure supplement 1A–B*). From this, we speculated that the 'Rab2 class' of shared phenotypes was in each case a result of a block in autophagosome-lysosome fusion. To explore this possibility, we tested a requirement for the homotypic fusion and protein sorting (HOPS) complex that is known to mediate SNARE-dependent autophagosome-lysosome fusion (*Jiang et al., 2014*; *Takáts et al., 2014*). As expected, disruption of the HOPS complex with RNAi of subunits, *Vps39*, *Vps18*, and *Vps11* each phenocopied the specific Rab2 class of IOM defects (*Figure 3E* and *Figure 3—figure supplement 1C*). These results suggest that a defect in autophagosome-lysosome fusion leads to the unique Rab2 class of RNAi phenotypes in IOM remodeling and predicts a novel role for Rab2 in autophagy.

## Rab2 or Rab7 knockdown blocks autophagosome-lysosome fusion with autophagy-dependent myofiber remodeling

To address the underlying role for the 'Rab2 class' of genes in T-tubule remodeling, we characterized autophagy in IOMs with *Rab2*, *Rab7* or *Stx17* knockdown at 4d APF. The mCherry:GFP:Atg8a autophagic flux reporter indicates both dually-labeled Atg8-marked autophagosomes and, due to the greater resistance of mCherry than GFP to lysosomal proteases, the successful delivery of autophagosomes to the lysosome by mCherry-labeled autolysosomes (*Kimura et al., 2007*). In control IOMs, there were dual mCherry-GFP-positive autophagosomes, as well as just mCherry-positive degradative autolysosomes (*Figure 4A*). In contrast, the *Rab2*, *Rab7* and *Stx17* RNAi IOMs contained a striking increase in dual mCherry-GFP-positive puncta (*Figure 4A–B*), showing that autophagosome clearance was severely blocked either due to accumulation of autophagosomes or nondegradative autolysosomes. The Stx17 SNARE localizes to the outer membrane of fully formed autophagosomes then detaches upon lysosomal degradation of the autophagosomal inner membrane (*Itakura et al., 2012*; *Tsuboyama et al., 2016*). In the RNAi conditions, Stx17 localized to the vesicle membranes (*Figure 4C*) also marked with Atg8 (*Figure 4D*; 0.53 Pearson correlation), indicating their identity as primarily mature autophagosomes. Confirming these results, TEM myofiber transverse sections (*Figure 2B*) revealed *Rab2*, *Rab7* or *Stx17* RNAi-depleted IOMs similarly and uniformly filled with thousands of accumulated autophagosomes carrying nondegraded cytoplasm and organelles

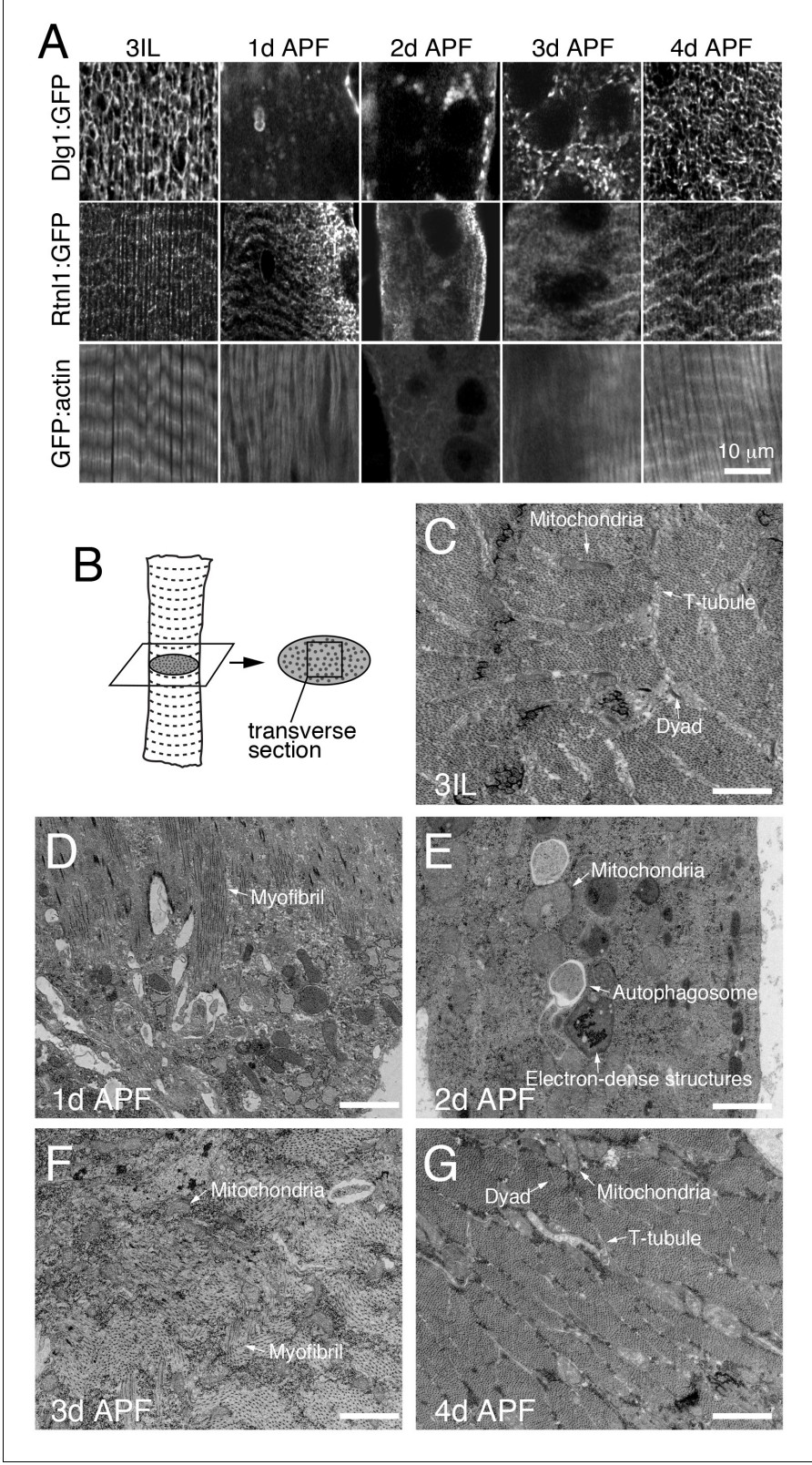

**Figure 2.** T-tubules disassemble and reassemble with IOM remodeling during metamorphosis. (**A**) Time course microscopy of Dlg1:GFP (T-tubule), Rtnl1:GFP (sarcoplasmic reticulum) or GFP:actin (myofibril) in wildtype animals at the indicated time points. (**B**) Schematic of an IOM TEM transverse section, as shown in 2**C**–**G**. (**C**–**G**) TEM images of IOM transverse sections in wildtype animals. Organized myofibrils and T-tubules were observed in both

*Figure 2 continued on next page*

*Figure 2 continued*

3IL and 4d APF stages (**C** and **G**). At 1d APF, myofibrils were partially lost with mostly disorganized membranes (**D**). At 2d APF, myofibrils were completely absent with obvious appearance of autophagosomes and electron-dense lysosomal compartments (**E**). At 3d APF, myofibrils were reassembled but not well organized with a lack of obvious T-tubules (**F**).

(*Figure 4E–F*). Thus, similar to previous reports of Rab7, Stx17, SNAP29 and Vamp7/8 functions in other Drosophila tissues (*Hegedűs et al., 2016*; *Takáts et al., 2013*), with myofiber remodeling, Rab2 also is required for autophagosome-lysosome fusion.

We further validated a requirement for autophagy in T-tubule remodeling by disrupting early steps in autophagy with muscle-targeted *Atg1*, *Atg3* or *Atg18* RNAi. In each case, RNAi resulted in disorganized and fragmented T-tubules and sparse myofibrils in IOMs at 4d APF (*Figure 5A–B*). Furthermore, TEM analysis of IOMs with *Atg1* or *Atg18* RNAi showed a lack of T-tubules, disorganized myofibrils and a striking accumulation of mitochondria that filled the cells (*Figure 5C*). The accumulation of fluorescently-marked mitochondria (*Figure 5D*) confirmed the TEM observations of free mitochondria within the cytoplasm with *Atg1* RNAi (*Figure 5C*) or inside mature but blocked autophagosomes with Rab2 RNAi (*Figure 4E*), suggesting that mitochondria are a major cargo of autophagy during IOM remodeling. As expected for a block in autophagy induction, and unlike the *Rab2* RNAi, autophagosomes did not occur or accumulate in IOMs with *Atg1* and *Atg18* RNAi.

## Autophagy is simultaneously upregulated with a requirement for proper T-tubule membrane disassembly

The autophagy requirement for T-tubule remodeling raised the questions whether levels of autophagy change and if required at distinct stages over IOM remodeling. In wildtype animals, we used imaging to monitor the amount of autophagosomes (GFP:Atg8) and endolysosomes/autolysosomes (GFP:Lamp1, GFP:Rab7) in live IOMs over 24h intervals of metamorphosis (*Figure 6A*). The number of Atg8 puncta indicative of autophagosomes dramatically increased by 1d APF and then sharply decreased by 3d APF (*Figure 6A*, top row and 6B). We confirmed a similar autophagosome distribution detected by immunostaining of endogenous Atg8 in wildtype IOMs at 1d APF (*Figure 6—figure supplement 1A*). These results parallel the increased prevalence of uniformly sized mCD8:GFP-marked vesicles (*Figure 1F*) and autophagosomes detected by TEM (*Figure 2D–F*) over 1-3d APF during wildtype IOM remodeling. Likewise, the number of Lamp1 and Rab7 marked endolysosomes or amphisomes/autolysosomes also increased during pupal stages 1d-3d APF (*Figure 6A*). Importantly, the increase in autophagy at 1d APF does not reflect an autophagy role in myofiber cell death, as the IOMs normally survive throughout metamorphosis (*Figures 1–2*) and assist days later with adult eclosion (*Ribeiro et al., 2011*).

The autophagy profile 1-3d APF (*Figure 6A*) appeared to inversely correlate with the presence of T-tubule membranes (*Figure 2*). To better determine the timing and relationship between autophagy and T-tubule disassembly, we performed a more focused timecourse experiment spanning IOM remodeling between late larval to 1d APF stages. We quantified a reproducible onset of autophagy induction between 12h and 18h APF that peaked by 20h APF in IOMs (*Figure 6C*, top row and 6D, red). In parallel experiments, we used live imaging of a $PI(4,5)P_2$ biosensor to detect T-tubules and the plasma membrane. While the T-tubules remained intact from third instar larval through 12h APF stages, the $PI(4,5)P_2$-marked T-tubule membrane network became partially fragmented by 18h APF and was completely absent by 20h APF in wildtype IOMs (*Figure 6C*, bottom row and 6D, blue). Only transient $PI(4,5)P_2$-marked membrane rings that appeared with T-tubule fragmentation could be detected in the central myofiber region up to 1d APF, while $PI(4,5)P_2$ continued to mark the plasma membrane throughout IOM remodeling. These results show highly reproducible, coincident onsets and peak occurrences of both T-tubule disassembly and autophagy induction between 18-20h APF (*Figure 6C–D*), supporting interrelated roles for autophagy and T-tubule remodeling.

Consistent with the timing of autophagy induction in wildtype IOMs (*Figure 6D*), a timecourse analysis of T-tubule remodeling in both the Rab2 class (*Rab2*, *Rab7* or *Stx17*) and Atg1 class (*Atg1*, *Atg18* RNAi) of RNAi phenotypes revealed that defects arose within the first day of IOM remodeling (*Figure 6E* and *Figure 6—figure supplement 2A–2B*). In all cases, normal T-tubule networks were

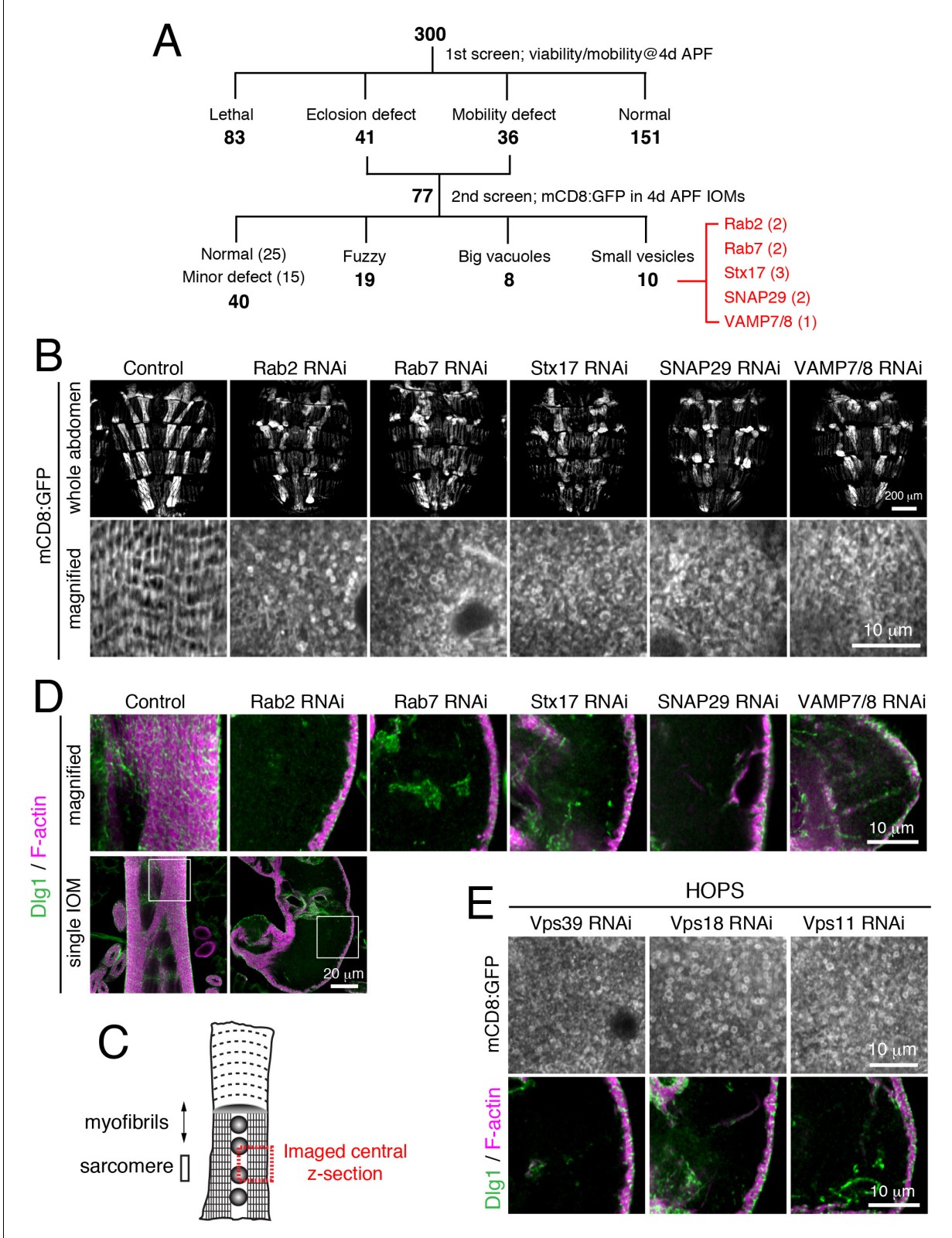

**Figure 3.** A unique T-tubule remodeling phenotype with knockdown of a set of known and unknown gene functions in autophagy. All IOMs imaged at 4d APF. (**A**) Muscle-targeted RNAi screen of IOM remodeling. In primary screen of 300 selected muscle-targeted RNAi lines (see text), 77 lines exhibited eclosion or adult mobility defects; these lines were used in a secondary screen for mCD8:GFP organization by confocal imaging. Three abnormal phenotype categories were identified for 37 lines. The shared 'small vesicle' phenotype was found for 10 RNAi lines for five genes presented

*Figure 3 continued*

here. (B) *Rab2*, *Rab7*, *Stx17*, *SNAP29* or *Vamp7/8* RNAi resulted in IOMs filled with small, mCD8:GFP-marked vesicles. Top row, brightly-marked dorsal IOMs in whole abdomen. Bottom row, magnified image of mCD8:GFP in single IOM. (C) Schematic of IOM and regions imaged in panel D–E. (D) T-tubule (Dlg1, green) and myofibril (F-actin, pink) organization in IOMs from control and *Rab2*, *Rab7*, *Stx17*, *SNAP29*, or *Vamp7/8* RNAi conditions. (E) RNAi of HOPS components, *Vps39*, *Vps18*, or *Vps11*, exhibited shared phenotypes of (top) many mCD8:GFP-marked small vesicles and (bottom) lack of T-tubules (Dlg1, green) and myofibrils (F-actin, pink). See *Figure 3—figure supplement 1* for related data.
The following source data and figure supplement are available for figure 3:

**Source data 1.** Relates to *Figure 3—figure supplement 1B*.
**Figure supplement 1.** A unique T-tubule remodeling phenotype identifies gene functions in autophagosome fusion.

observed in the IOM precursor muscles at the third instar larval stage (*Figure 6E*, left), in line with the normal larval mobility observed with each muscle-targeted RNAi depletion. However, initial defects in mCD8:GFP organization appeared between 12h and 18h APF (*Figure 6E*), at a time when normally both T-tubule disassembly and autophagy induction occur (*Figure 6C–D*). These knockdown IOMs remained persistently blocked at an 18–20h stage in remodeling throughout metamorphosis.

In control IOMs at the time of T-tubule disassembly at 18–20h APF (*Figure 6C–6D*), mCD8:GFP-marked membranes reorganized into dispersed bright patches and some vesicles (*Figure 6E*, top row and *Figure 6F*). In the case of the Rab2 class of RNAi phenotypes, T-tubule disassembly at 18–24h APF occurred in conjunction with an abnormal accumulation of uniformly-sized mCD8:GFP vesicles (*Figure 6E*, middle row; *Figure 6F*; *Figure 6—figure supplement 2B*, top rows), many of which were identified as Atg8-marked autophagosomes (*Figure 6G*). In contrast, the Atg1 class of RNAi phenotypes showed normal timing in initial T-tubule membrane disassembly at 18–20h APF, but with abnormally large clusters of bright mCD8:GFP-positive membrane patches (*Figure 6F*). By 1d APF, the bright membrane clusters increased in size to reveal stacked membrane whirls that persisted to 4d APF (*Figure 6E*, bottom row; *Figure 6—figure supplement 2B*, bottom row; *Figure 5A*). Together with the lack of T-tubules and autophagosomes in IOMs with autophagy disruption at 4d APF (*Figure 5*), these results indicate that autophagy plays a key role downstream of initiation and membrane scission in T-tubule disassembly for subsequent mobilization of disassembled T-tubule-derived membranes.

While the specific phenotypes were distinct with knockdown of genes required either for autophagy induction/autophagosome biogenesis (*Atg1*, *Atg3* and *Atg18*) or autophagy clearance (*Rab2*, *Rab7* and *Stx17*), all autophagy functions were consistently required by 18–20h APF when both autophagy induction and T-tubule disassembly normally occurs. Thus, in IOMs, changes in autophagy levels are concurrently regulated and required for myofiber T-tubule membrane remodeling, and specifically, the *Atg1* pathway mediates autophagy-dependent progression of T-tubule membrane disassembly.

## Rab2 is required for autophagic flux in other tissues and in mammalian cells

To generalize our findings to other cell types, we tested effects of Rab2 RNAi in third instar larval fat body, a tissue with a well-established starvation-induced autophagy response. Consistent with our data in IOMs, Rab2 RNAi increased the amount of both Atg8-marked autophagosomes and uncleared p62 autophagic cargo (*Figure 7—figure supplement 1A–B*) and blocked mCh:GFP:Atg8a autophagic flux (*Figure 7—figure supplement 1C–D*). Unlike in myofibers, TEM analysis of Rab2-depleted fat body indicated the site of accumulated autophagic cargo at late stages of autophagy with an increased number and size of amphisomes or autolysosomes, often with nondegraded cytoplasmic contents (*Figure 7—figure supplement 1E–F*). Altogether, our results show that Rab2 plays important functions at late stages of autophagy broadly across Drosophila tissues, although with different sensitivities to knockdown and/or specific requirements for autophagosome or amphisome delivery to lysosomes.

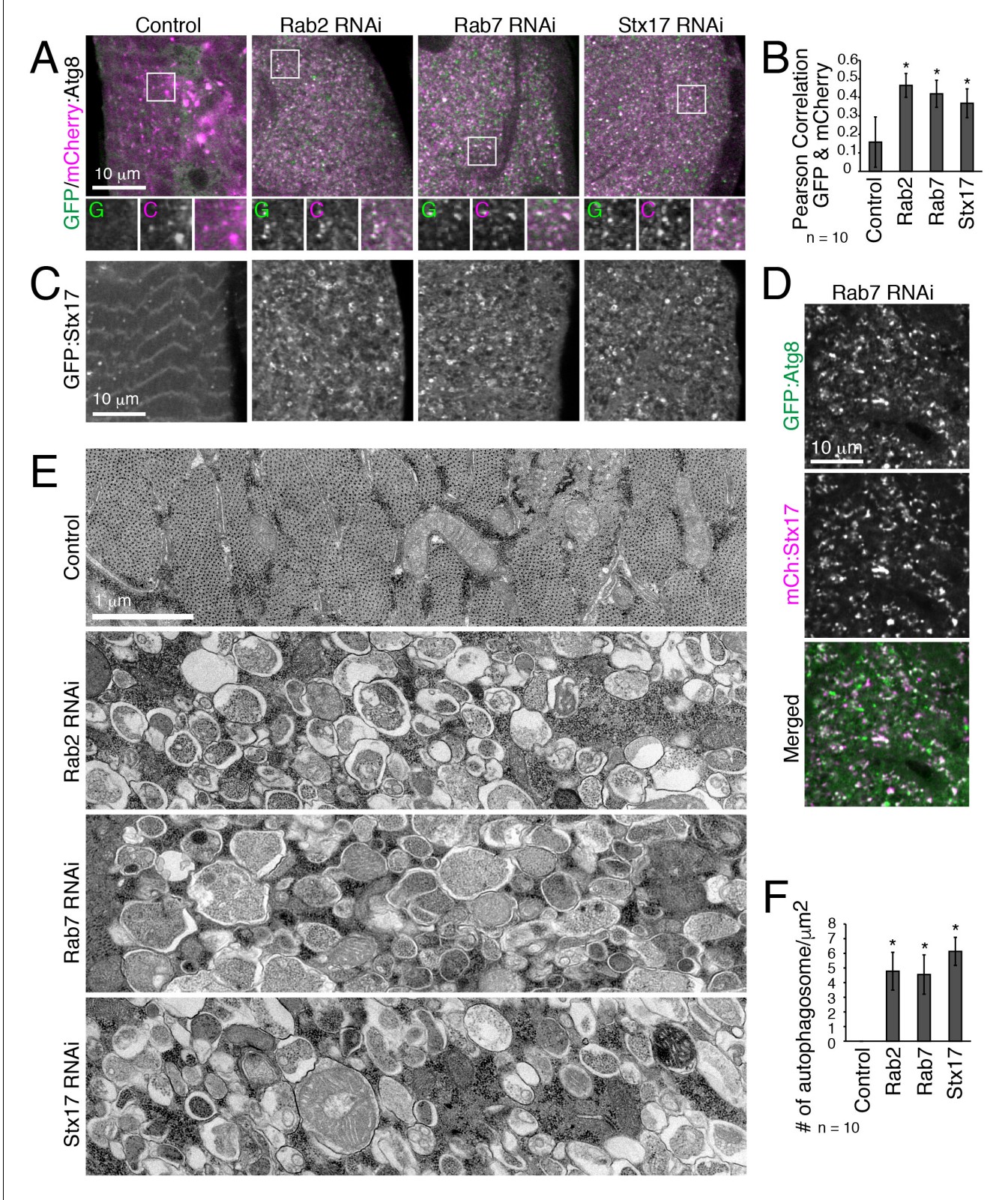

**Figure 4.** Autophagosomes accumulate in IOMs with *Rab2, Rab7* or *Stx17* knockdown. All IOMs imaged at 4d APF. (**A**) Autophagic flux assay using tandem-tagged mCherry:GFP:Atg8 (mCherry (C), pink; GFP (G), green; colocalization, white). Peripheral IOM z-sections with magnified regions from indicated boxed areas shown below. In control IOMs, mCherry-positive only puncta were primarily detected, indicative of Atg8 flux to autolysosomes. In *Rab2*, *Rab7* or *Stx17* RNAi IOMs, dual-positive Atg8 puncta were primarily detected, indicating block in autophagic flux. (**B**) Pearson correlation

*Figure 4 continued on next page*

Figure 4 continued

between GFP and mCherry of mCherry:GFP:Atg8 from pooled data for 10 images from three experiments, ± SD. (C) GFP:Stx17 distribution in IOMs from control or with *Rab2*, *Rab7* or *SNAP29* RNAi, which show increased GFP:Stx17 localization at puncta and small rings. (D) Colocalization of GFP: Atg8 and mCherry:Stx17 (Pearson correlation, 0.53) in *Rab7* RNAi IOMs. (E–F) TEM images of IOM transverse-sections. Control IOMs show expected myofibrils and T-tubule membranes, while *Rab2*, *Rab7* or *Stx17* RNAi IOMs were filled mostly with autophagosomes. (F) Quantification of the mean number of autophagosomes per IOM area, ± SD.
The following source data is available for figure 4:

**Source data 1.** Relates *Figure 4B*.
**Source data 2.** Relates to both *Figure 4F* and *Figure 5C*.

Mammals each have two Rab2 isoforms, Rab2A and 2B (*Aizawa and Fukuda, 2015*). To examine whether Rab2 has an evolutionarily conserved role in autophagy, we deleted Rab2A and Rab2B using the CRISPR/Cas9 system in mouse embryonic fibroblasts (MEFs). We generated both single knockouts (Rab2A_KO, Rab2B_KO) and a double knockout cell line (Rab2A/B_DKO; *Figure 7A*). In the 'fed' culture condition, the number of LC3 puncta indicative of autophagosomes significantly increased in Rab2A/B_DKO versus in control cells (*Figure 7B–C*). Bafilomycin A1 treatment elevated the level of LC3-II in parental MEFs but not in Rab2A/B_DKO MEFs (*Figure 7D*), discriminating that Rab2A/B_DKO blocks autophagy flux. Since single expression of either Rab2A or 2B rescued the phenotype of Rab2A/B_DKO in autophagy (*Figure 7—figure supplement 2A–B*), Rab2A/B_DKO MEFs were used for further analysis.

In Rab2A/B_DKO versus parental MEFs grown in fed conditions, TEM analysis revealed more amphisomes or autolysosomes, identified as electron-dense and single-membrane delimited compartments with cytoplasmic contents, but not of unfused autophagosomes (*Figure 7E–G* and *Figure 7—figure supplement 2C–D*), similar to Rab2 knockdown Drosophila fat body. This indicates that Rab2A/B are required for amphisome-lysosome fusion or autophagic clearance at autolysosomes, however, are not essential for autophagosome-lysosome fusion in MEFs. To test if the accumulation of autolysosomes results from a defect in lysosomal function, we used 'mosaic analysis' by mixing unmarked parental cells and stably GFP-marked Rab2A/B_DKO MEFs to compare effects of both genotypes side by side in the exact same condition. Consistent with a block in autophagy flux, Rab2A/B_DKO MEFs (GFP+) had more LC3 puncta than seen in parental MEFs (GFP-) grown in a mosaic culture (*Figure 7—figure supplement 2E*). There was no difference between parental and Rab2A/B_DKO MEFs in lysosome functions needed for degradation, as indicated by number and size of acidified compartments (*Figure 7H* and *Figure 7—figure supplement 2F*; LysoTracker) or by Cathepsin B activity (*Figure 7I* and *Figure 7—figure supplement 2G*; Magic Red). Furthermore, there was no difference between parental and Rab2A/B_DKO MEFs in the kinetics or levels of ligand-induced epidermal growth factor receptor (EGFR) degradation, an established indicator of lysosomal activity (*Figure 7—figure supplement 2H–I*). Thus, overall lysosomal maturation appears unaffected by Rab2A/B_DKO despite a block in autophagic clearance. Collectively, a conserved Rab2 function is indispensable for autophagic flux, which for Rab2A/B is required after autophagosome delivery in MEFs.

## Rab2 localizes to mature autophagosomes

It is reported that Drosophila Rab2 has an affinity with the HOPS complex in both Drosophila and mammals (*Gillingham et al., 2014*; *Kajiho et al., 2016*). We confirmed the interaction using Rab2A/B co-immunoprecipitation assays. As previously reported, Rab7 but not Rab5 had an affinity with Vps39 and Vps41, two selective Rab-binding components of the HOPS complex (*Figure 8A*). Similar to Rab7, both Rab2A and Rab2B were also co-immunoprecipitated with Vps39 and Vps41 (*Figure 8A*). The HOPS complex tethers membrane compartments to promote fusion between endosomes and lysosomes (*Balderhaar and Ungermann, 2013*) or between autophagosomes and lysosomes (*Jiang et al., 2014*; *Takáts et al., 2014*), and Rab7 has been localized to late endosomes/lysosomes in both fusion events (*Hyttinen et al., 2013*).

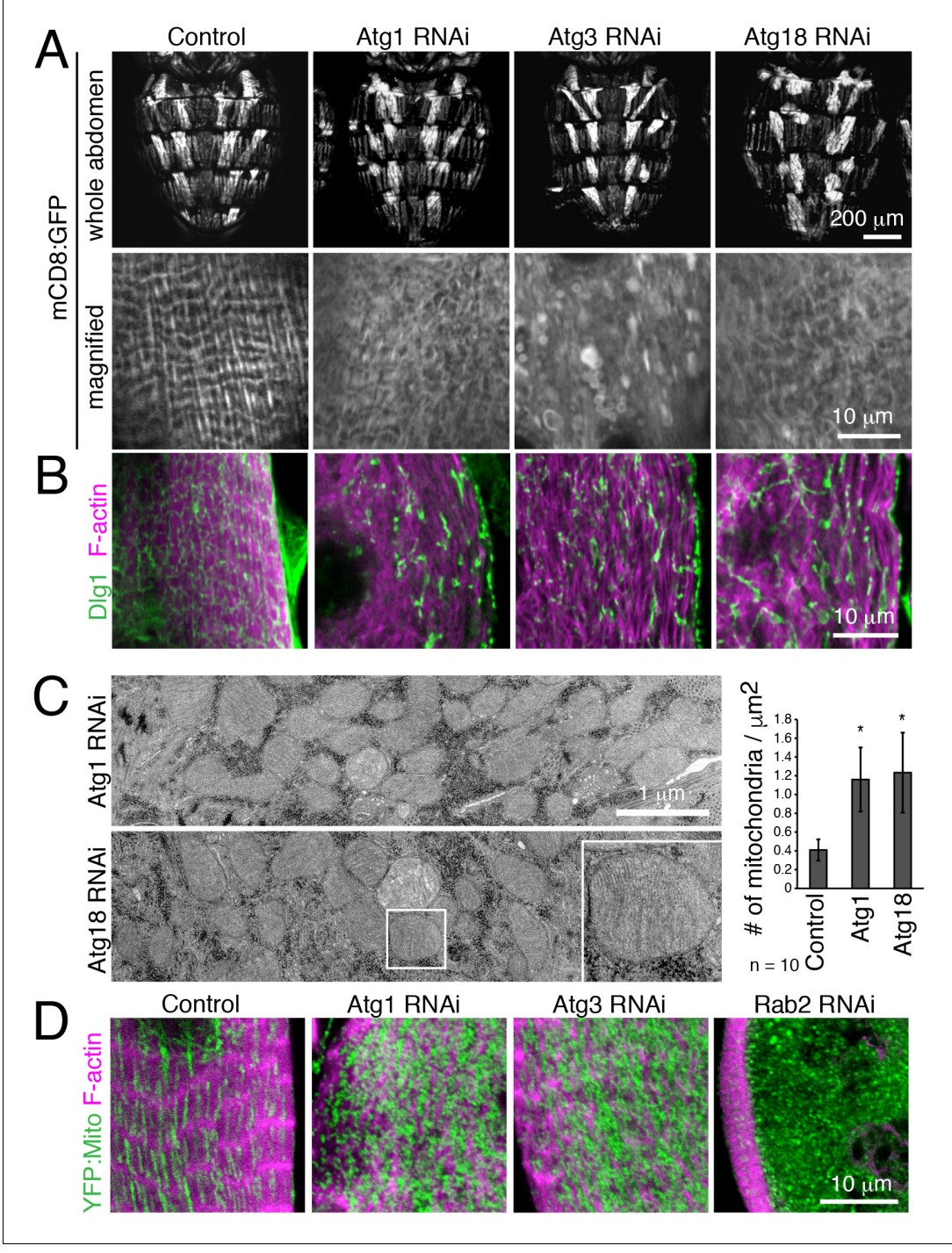

**Figure 5.** Autophagy is required for IOM T-tubule remodeling and mitochondrial clearance. (**A**) mCD8:GFP in 4d APF dorsal abdominal muscles (top) and IOM section (bottom) for control and *Atg1*, *Atg3* or *Atg18* RNAi conditions. (**B**) T-tubule (Dlg1, green) and myofibril (F-actin, pink) organization in IOMs of control and *Atg1*, *Atg3* or *Atg18* RNAi that show fragmented and disorganized T-tubules. (**C**) TEM images of IOM transverse-sections show disorganized contractile system, lack of T-tubules and many mitochondria in *Atg1* or *Atg18* RNAi conditions at 4d APF. Quantification of the mean number of mitochondria per area, ± SD. (**E**) Mitochondria (YFP:Mito, green) and myofibril (F-actin, pink) organization in control and *Atg1*, *Atg18* or *Rab2* RNAi IOMs at 4d APF.

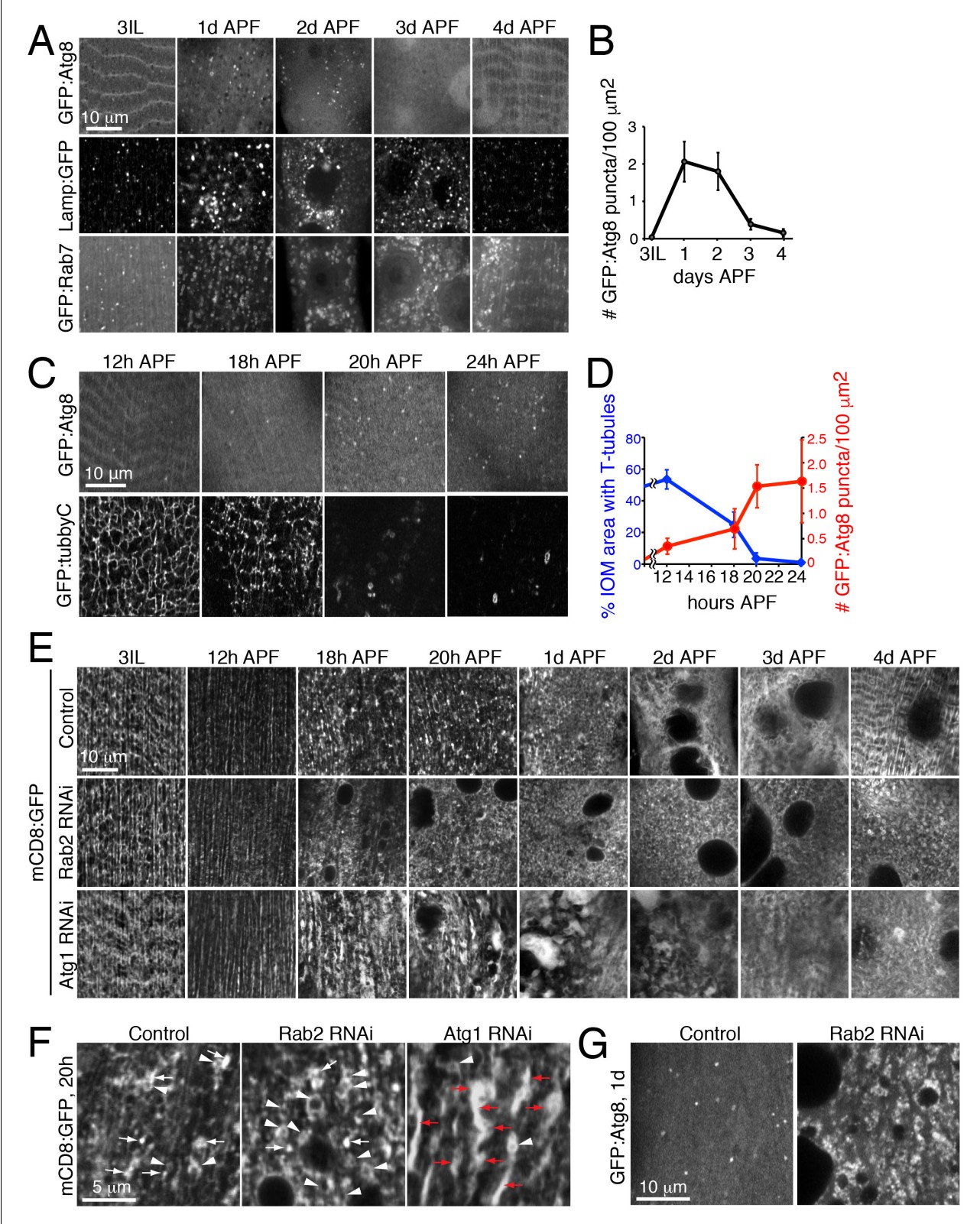

**Figure 6.** Autophagy induction is coincident with and required for proper T-tubule membrane disassembly. (A) Time course microscopy of autophagy-related markers in live wildtype animals over 1 day intervals during metamorphosis; GFP:Atg8 (autophagosomes), Lamp1:GFP (endolysosomes/autolysosomes), GFP:Rab7 (late endosomes/amphisomes). (B) Quantification of mean GFP:Atg8 puncta number per tissue area ± SD from at least 10 randomly selected IOMs. (C–D) Time course microscopy of GFP:Atg8 (autophagosomes) and GFP:Tubby-Cter [PI(4,5)P2, T-tubules] in live wildtype

*Figure 6 continued on next page*

*Figure 6 continued*

animals at the indicated hours APF (**C**). Mean percentages IOM area covered by T-tubules (blue, ± SD) and mean number of autophagosomes (red, ± SD) quantified from at least 10 randomly selected IOMs. (**E**) Time course microscopy during metamorphosis of mCD8:GFP in IOMs of control, *Rab2* RNAi or *Atg1* RNAi (as in **Figure 1F**, bottom). Both *Rab2* and *Atg1* RNAi show normal T-tubules in 3IL muscle and initial defects in membrane organization by 1d APF that persist as a block in remodeling through 4d APF. (**F**) mCD8:GFP membrane in central regions of IOMs at 20h APF upon T-tubule disassembly in control, *Rab2* RNAi or *Atg1* RNAi, with mCD8:GFP membrane reorganization into bright patches (arrows), big membrane patches/stacks (red arrows) or membrane vesicles (arrowheads). (**G**) GFP:Atg8 at 1d APF in control and *Rab2* RNAi IOMs. See **Figure 6—figure supplements 1** and **2** for related data.

The following source data and figure supplements are available for figure 6:

**Source data 1.** Relates to *Figure 6B*.
**Source data 2.** Relates to *Figure 6D*.
**Source data 3.** Relates to *Figure 6D*.
**Figure supplement 1.** Characterization of autophagy levels during IOM remodeling.
**Figure supplement 2.** Characterization of autophagy requirement for IOM membrane remodeling.

We investigated Rab2 localization during autophagy in both fly IOMs and in MEFs. Whereas in wildtype IOMs at 4d APF relatively few YFP:Rab2 puncta were present, we observed an accumulation of YFP:Rab2 puncta and uniformly-sized rings with the accumulation of autophagosomes in *Rab7* or *Stx17* RNAi conditions (**Figure 8B**). Colocalization between YFP:Rab2 and mCherry:Atg8 confirmed autophagosome identity of Rab2-marked compartments in *Rab7* RNAi IOMs (**Figure 8C–D**). These results indicate that Rab2 localizes to autophagosomes in fly IOMs. Similarly in MEFs, both Rab2A and Rab2B significantly colocalized with LC3 (**Figure 8E–F** and **Figure 8—figure supplement 1A**). Immuno-EM also identified colocalization of endogenous LC3 and GFP:Rab2A on the same membranes of autophagic structures with the typical appearance of autophagosomes containing undigested contents (**Figure 8—figure supplement 1B**). It is established that LC3 localizes to mature autophagosomes as well as the isolation membrane, an elongating pre-autophagosome structure (**Kabeya et al., 2000**; **Mizushima et al., 2001**) marked by Atg16L1 (**Mizushima et al., 2003**). To assess to what extent Rab2A/B localizes to the isolation membranes and/or autophagosomes, we examined colocalization between Atg16L1, LC3 and GFP:Rab2A/B. Rab2A/B colocalized well with LC3 but not noticeably with Atg16L1 (**Figure 8G–H** and **Figure 8—figure supplement 1C–D**). Rab2A/B also did not colocalize with Lamp1, a late endosome-lysosome marker (**Figure 8—figure supplement 1E–H**). Colocalization analysis among Rab2A, Rab7 and LC3 or Lamp1 revealed that Rab2A and Rab7 preferentially localized to autophagosomes and endolysosomes, respectively (**Figure 8—figure supplement 1I–L**). Altogether, the above results show that fly and mouse Rab2 specifically localizes to completed autophagosomes.

To characterize the dynamics of Rab2A localization in autophagy, we performed live cell time-lapse imaging. In starved MEFs, recruitment of CFP:LC3 preceded YFP:Rab2 by several minutes (**Figure 9—figure supplement 1A**), consistent with Rab2A localization to autophagosomes but not isolation membranes (**Figure 8G–H**). In turn, recruitment of YFP:Rab2A to autophagosomes preceded CFP:Stx17 by several minutes in most cases (**Figure 9A**), suggesting that Rab2A localization is independent of Stx17. Consistent with the timing of Rab2A recruitment, Drosophila Rab2 localization at autophagosomes was unaffected by *Rab7*, *Stx17* or *Vps39* RNAi in IOMs (**Figure 9B–C**). Since Rab2 has affinity with the HOPS complex, we tested whether *Rab2* RNAi affects the recruitment of HOPS to autophagosomes. As previously reported (**Takáts et al., 2014**), Dor, a fly homolog of the Vps18 HOPS complex component, colocalized with GFP:Atg8 in control IOMs (**Figure 9D**). In contrast, this colocalization was diminished by *Rab2* RNAi (**Figure 9E**), indicating that Rab2 is important for recruitment of the HOPS complex to autophagosomes. Collectively, Rab2 is an autophagosome-associated Rab GTPase that promotes autophagosome-lysosome fusion through interactions with the HOPS complex (**Figure 9F**), and T-tubule remodeling requires a high volume of membrane flux through the autophagy pathway.

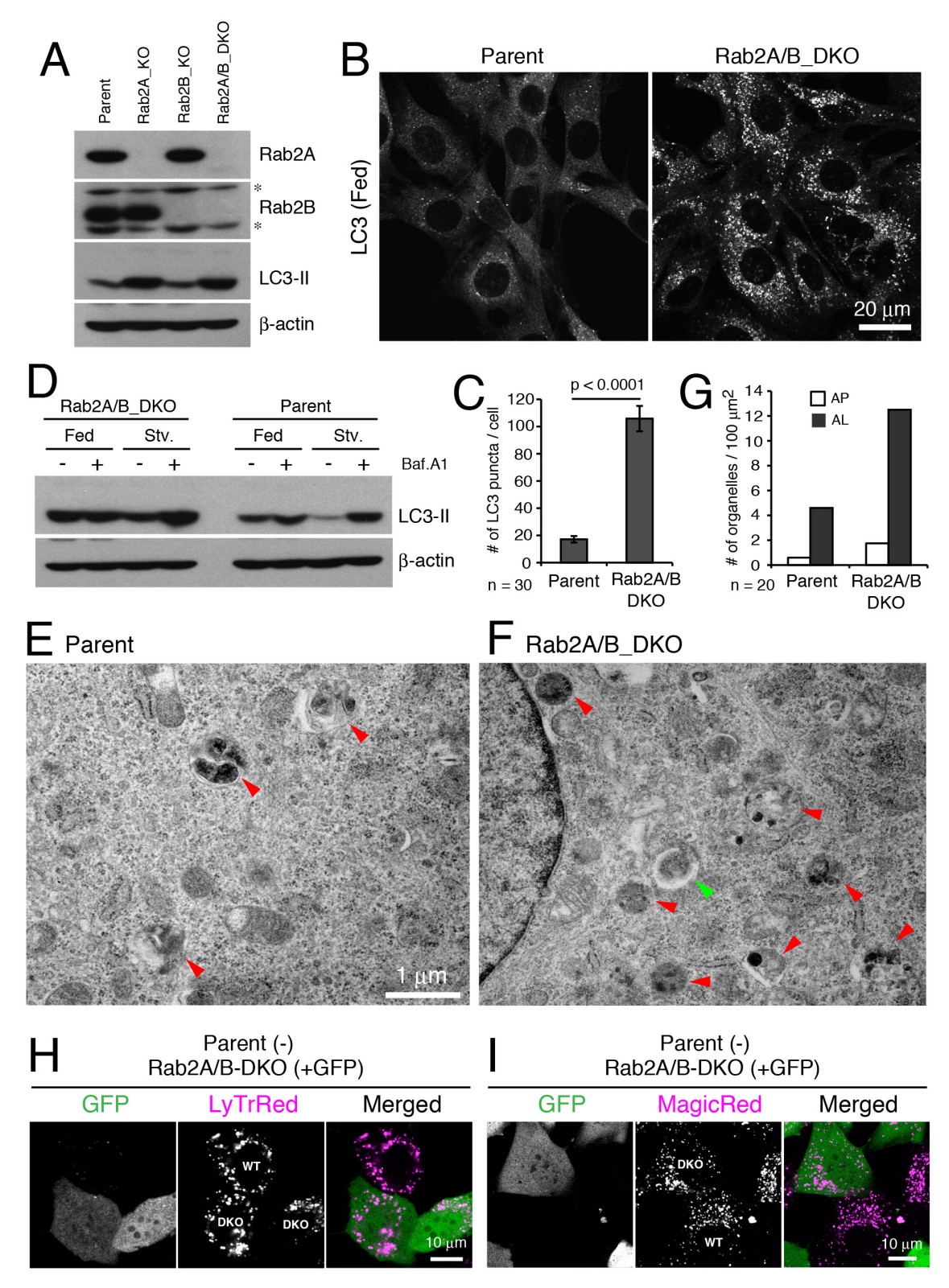

**Figure 7.** Rab2 has a conserved function required for autophagic clearance in MEFs. (A) Rab2A/B double knockout (DKO) MEFs generated by CRISPR-Cas9. Parental, Rab2A-KO, Rab2B-KO, or Rab2A/B-DKO MEFs were cultured in regular growth medium and lysates were analyzed by immunoblots with indicated antibodies. Asterisks denote non-specific bands. (B–C) LC3 puncta in Rab2A/B_DKO MEFs. Parental or Rab2A/B_DKO MEFs were cultured in regular medium and analyzed by immunofluorescence microscopy (B). Mean number of LC3 puncta from 30 cells ± SEM of pooled data from three

*Figure 7 continued on next page*

*Figure 7 continued*

experiments (**C**). (**D**) LC3-II flux assay in parental or Rab2A/B_DKO. Parental or Rab2A/B_DKO MEFs were cultured in regular medium (Fed) or EBSS (Stv) for 2h with or without 100 nM Bafilomycin A1 and analyzed by immunoblotting. (**E–G**) TEM analysis of parental (**E**) or Rab2A/B_DKO MEFs (**F**) cultured in regular medium. Autolysosomes, red arrowheads; Autophagosomes, green arrowhead. Quantification of the number of autophagosomes or autolysosomes from 20 randomly selected areas (**G**). (**H–I**) Mosaic analysis of parental (GFP-) and Rab2A/B_DKO MEFs (GFP+) with Lysotracker Red (**H**) or Magic Red (**I**). See *Figure 7—figure supplements 1* and *2* for related data.

The following source data and figure supplements are available for figure 7:

**Source data 1.** Relates to *Figure 7C*.
**Source data 2.** Relates to *Figure 7G*.
**Source data 3.** Relates to *Figure 7—figure supplement 1B and D*.
**Source data 4.** Relates to *Figure 7—figure supplement 1F*.
**Source data 5.** Relates to *Figure 7—figure supplement 2I*.
**Figure supplement 1.** *Rab2* RNAi blocks autophagic flux in third instar larval fat body.
**Figure supplement 2.** Characterization of Rab2A/B double knockout MEFs.

## Discussion

The importance of T-tubule organization for muscle function is well established. However, the dynamics and mechanisms that shape the T-tubule membrane network are largely unexplored, in part due to methodological challenges in many systems. In the present study, we characterized a wildtype myofiber remodeling program by confocal and electron microscopy in intact muscles in vivo. In Drosophila IOMs during metamorphosis, the entire contractile and excitation-contraction coupling system, including T-tubules, are disassembled and then reassembled. This process high-lights that myofibers harbor distinct programs for initial T-tubule formation versus regulated T-tubule remodeling. This likely includes additional mechanisms for T-tubule membrane disassembly and ren-ovation, features that reflect those seen with mammalian myofiber atrophy and recovery (*Piccirillo et al., 2014*). We demonstrate that the Drosophila body wall muscles provide an unprece-dented system permitting a combination of powerful visualization and systematic perturbation analy-sis, including the first genetic screens, of T-tubule dynamics and organization. We uncovered both that autophagy is indispensable for the remodeling, and that Rab2 plays an unexpected role at auto-phagosomes for autophagic clearance in this process and in broader contexts.

Autophagy is upregulated with the onset of IOM remodeling during metamorphosis (*Figure 6A–D*). Further, disruption of autophagy initiation, autophagosome formation or clearance all induced loss of T-tubules with a block in IOM remodeling at/after T-tubule disassembly (*Figure 6E*). As far as we know, this is the first report of a non-cell death role of autophagy in Drosophila metamorphosis. The role of autophagy in IOMs that persist and redifferentiate during metamorphosis is clearly differ-ent from its roles in pupal midgut and salivary gland cells that undergo autophagic forms of cell death (*Berry and Baehrecke, 2007*; *Chang et al., 2013*; *Das et al., 2012*; *Lee et al., 2002a*). There are multiple speculative direct or indirect role(s) for autophagy specifically in T-tubule membrane remodeling: (1) a direct role in T-tubule membrane recycling, as a means to deliver disassembled T-tubule membrane via autophagosomes to lysosomes or related organelles for intracellular storage, then later redeployed to contribute to T-tubule reassembly; (2) an indirect role in cell renovation, including T-tubule membrane clearance, to permit cell space for redifferentiation; or (3) an indirect role in cell metabolism, to support cell survival and/or the energy cost of redifferentiation with star-vation during metamorphosis. Most likely, autophagy serves some combination of these roles in IOM remodeling.

How could autophagy play a direct role in T-tubule remodeling? It was surprising that mCD8: GFP-positive small vesicles accumulated to a similar degree as autophagosome numbers in IOMs

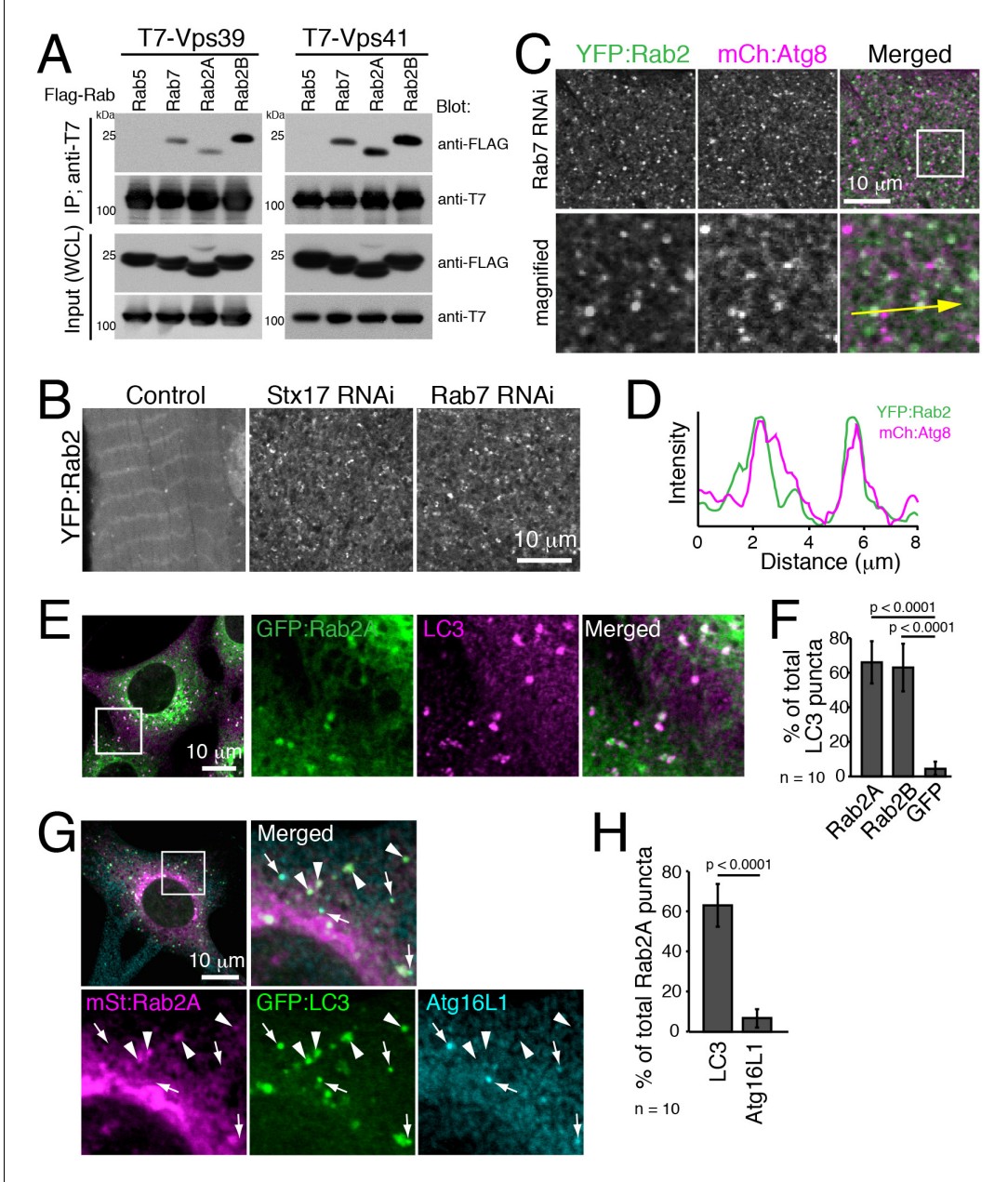

**Figure 8.** Rab2 localizes to completed autophagosomes. (**A**) Interaction between Rab2A/B and Vps39 or Vps41. COS-7 cells were co-transfected with T7-Vps39 or -Vps41 and FLAG-tagged Rab as indicated. Two days later, cells were lysed, immunoprecipitated with anti-T7 antibody, and detected by immunoblots as indicated. (**B**) YFP:Rab2 distribution in IOMs at 4d APF in control, *Stx17* or *Rab7* RNAi conditions. The number of YFP:Rab2 puncta and rings increased when autophagosome-lysosome fusion was blocked. (**C–D**) Colocalization between YFP:Rab2 and mCherry:Atg8 in *Rab7* RNAi IOMs at 4d APF (**C**). Line plot profile of a yellow arrow in panel C (**D**). (**E–F**) Colocalization between Rab2A and LC3 in MEFs. MEFs stably-expressing GFP:Rab2A were cultured in EBSS for 1h and immunostained for LC3 (**E**). Mean percent colocalization with LC3 ± SD of 10 images (**F**). (**G–H**) Colocalization analysis between stably-expressed mStrawberry:Rab2A, GFP:LC3 and Atg16L1 in MEFs cultured in EBSS for 1h. Arrows: LC3-positive, Atg16L1-positive isolation membranes. Arrowheads: LC3-positive, Atg16L1-negative autophagosomes (**G**). Mean percent colocalization with Rab2A ± SD of 10 images (**H**). See *Figure 8—figure supplement 1* for related data.

The following source data and figure supplement are available for figure 8:

**Source data 1.** Relates to *Figure 8F*.
**Source data 2.** Relates to *Figure 8H* and *Figure 8—figure supplement 1D*.

*Figure 8 continued on next page*

*Figure 8 continued*

**Source data 3.** Relates to *Figure 8—figure supplement 1F and H*.

**Source data 4.** Relates to *Figure 8—figure supplement 1J and L*.

**Figure supplement 1.** Rab2A and Rab2B localize to complete autophagosomes, but not isolation membranes nor lysosomes.

when autophagosome-lysosome fusion was blocked (*Figures 3* and *4*). This suggests that mCD8: GFP localizes to autophagosomes during IOM remodeling. It is possible that T-tubule membranes are a source of autophagosomal membrane, at least in part: mCD8:GFP labels the muscle plasma membrane and T-tubules in larval muscle precursor cells of IOMs (*Figure 1F*), and T-tubule disassembly coincides with the upregulation in autophagy early in metamorphosis (*Figure 6C–D*). Also, disruption of autophagy induction blocked normal progression in disassembly and remodeling of T-tubule-derived mCD8:GFP-marked membranes (*Figure 6E* and *Figure 6—figure supplement 2B*). In the absence of autophagy initiation, mCD8:GFP-positive stacked membranes were observed (*Figure 6E* and *Figure 6—figure supplement 2B*), likely retained or partially disassembled T-tubules. We propose that T-tubules are remodeled through autophagosomes. It is important to note that T-tubules are not an apparent autophagic cargo, but instead, a possible source of autophagosome membrane. In this scenario, T-tubules are disassembled into autophagosomes and then reassembled from subsequent autolysosome-related structures (*Figure 9F*), both of which successively increased in numbers during wildtype IOM remodeling (*Figure 6A*).

Alternatively or additionally, other roles for autophagy could indirectly impact T-tubule remodeling. Extensive IOM atrophy with nearly complete disassembly of the contractile and excitation-contraction systems by 1d APF is followed by a rapid re-differentiation within hours after 3.5d APF. Autophagy could be required to simply clear away and degrade the old contraction systems in order to make space to rebuild and realign new systems, as well as permit the normal central repositioning of nuclei away from the myofiber cortex. However, the persistent block in early IOM remodeling with autophagy disruption suggests that the remodeling normally proceeds through a progression of interrelated steps rather than independent programs for disassembly and reassembly. Autophagy also has a well-established role in metabolic homeostasis through the recycling of amino acids and turnover of damaged mitochondria in the lysosome. Our data suggests that mitochondria are a major autophagic cargo with IOM remodeling. In conditions that disrupted autophagy initiation (*Atg1, Atg18* RNAi), the cytoplasm was abnormally filled with mitochondria in IOMs at 4d APF (*Figure 5C*). Consistent with that, a significant portion of autophagosomes harbored intact mitochondria when autophagosome-lysosome fusion was blocked (*Rab2, Rab7 or Stx17 RNAi*; *Figure 4E*). This is different from observations in larval muscle, in which mitochondria were notably absent in autophagosomes that accumulated with a block in autophagy (*Zirin et al., 2013*). It is possible that mitophagy, a selective form of autophagy for mitochondrial turnover, is upregulated and could play both metabolic and cell renovation roles in IOM remodeling. Interestingly, the autophagy-blocked IOMs remained viable throughout metamorphosis, suggesting that autophagy is not absolutely required for cell survival through the starvation with metamorphosis.

Through a systemic screen of all Drosophila Rab GTPases, we uncovered an unexpected role for Rab2 in autophagy. The striking Rab2 RNAi IOM phenotype was shared with RNAi of other functions known to be specifically required for autophagosome-lysosome fusion. Genetic blockade of autophagosome-lysosome fusion resulted in a dramatic phenotype, with massive accumulations of autophagosomes within IOMs (*Figure 4*). Previously, autophagosome-lysosome fusion was shown to involve the cooperative functions of Rab7, the HOPS tethering complex, and a trans-SNARE complex between Stx17, SNAP29 and VAMP7/8 (*Shen and Mizushima, 2014*). Among these tethering and fusion functions, it has been shown that Stx17 (a hairpin SNARE) is recruited to autophagosomal membranes, while Rab7 and VAMP7/8 localize to endolysosomal membranes. Stx17 localizes to autophagosomes as well as to the ER and mitochondria (*Itakura et al., 2012*), but the HOPS complex directly associates and colocalizes with Stx17 only at autophagosomes (*Jiang et al., 2014*; *Takáts et al., 2014*). This suggests that Stx17 is not a sole determinant for HOPS complex recruitment.

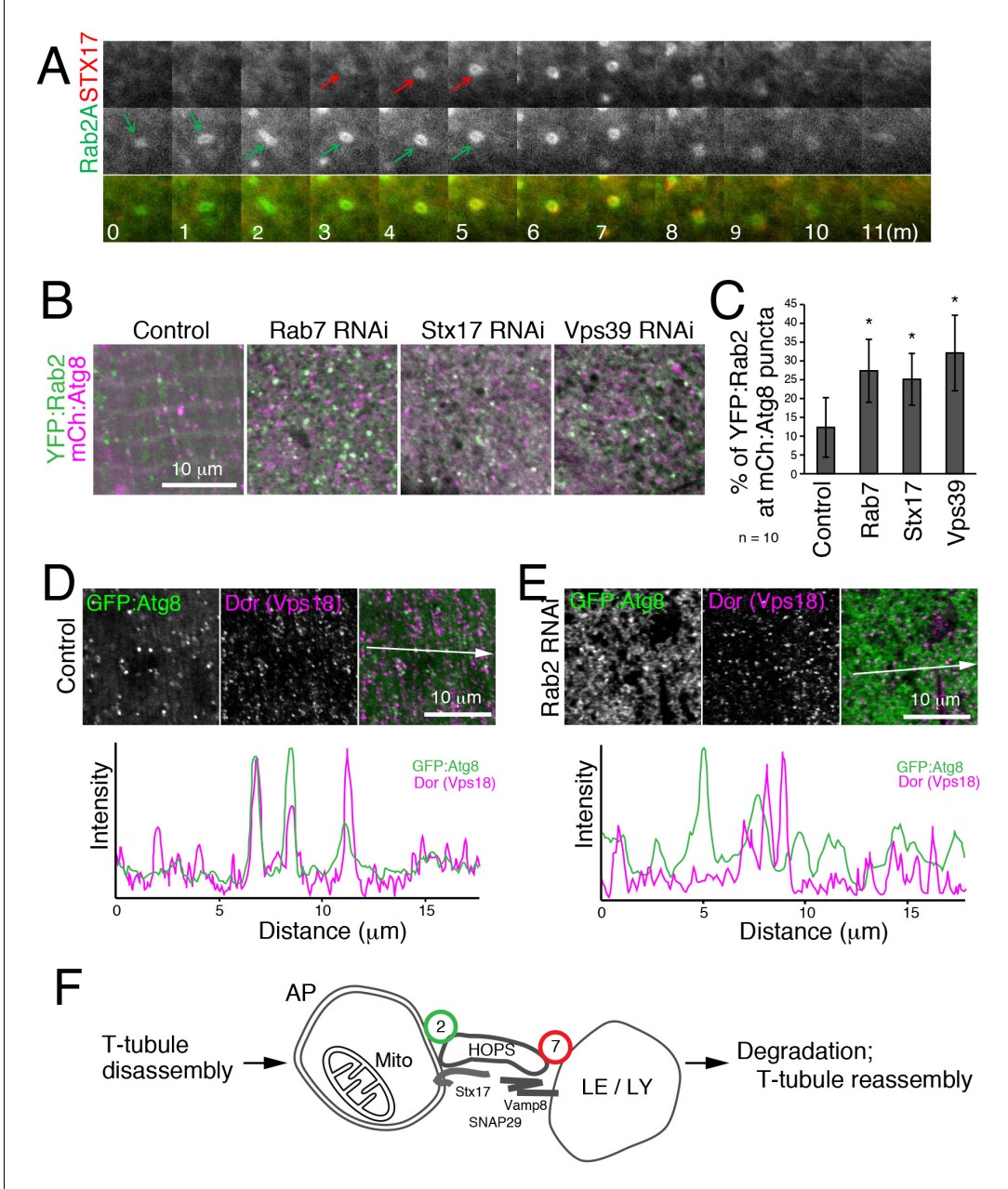

**Figure 9.** Hierarchal analysis of Rab2 and factors involved in autophagosome-lysosome fusion. (**A**) Dynamics of EYFP:Rab2A and super-enhanced CFP (seCFP):Stx17TM in autophagy. MEFs expressing both EYFP:Rab2A and seCFP:Stx17TM were incubated in amino acid free DMEM and imaged by fluorescent microscopy. Rab2A localization preceded Stx17 by several minutes in most cases. (**B–C**) Colocalization of YFP:Rab2 with mCherry:Atg8 in 4d APF IOMs with *Rab7*, *Stx17* or *Vps39* RNAi (**B**), indicating Rab2 localization to autophagosomes is independent of these functions. Mean percentage of YFP:Rab2-positive puncta colocalized with mCh:Atg8 puncta quantified from 10 randomly selected areas (**C**). (**D–E**) Colocalization between mCherry: Atg8 and Dor (Vps18) in 1d APF control IOMs (**D**) or Rab2 RNAi IOMs (**E**). Line plot profiles of Atg8 and Dor intensities along white arrows in merged images. (**F**) Model for autophagy-mediated T-tubule remodeling and a Rab2 role in autophagosome-lysosome fusion. In myofiber remodeling, progression from T-tubule membrane disassembly requires Atg1-mediated autophagy induction and may contribute as a source of autophagosomal membranes. Mitochondria are a major autophagic cargo in this process. Rab2 localizes to completed autophagosomes and interacts with the HOPS complex to promote autophagosome-lysosome fusion, leading to cargo degradation. The autolysosomal membranes could be recycled back as a source for T-tubule membrane reassembly or other membrane structures in the cell.

The following source data and figure supplement are available for figure 9:

**Source data 1.** Relates to *Figure 9C*.

*Figure 9 continued on next page*

*Figure 9 continued*

**Figure supplement 1.** Rab2 is recruited to mature autophagosomes and required for Dor/Vps18 HOPS localization

We propose that Rab2 is required for the autophagosomal recruitment of the HOPS complex. Rab2 specifically localized to completed autophagosomes (*Figures 8–9*), and Rab2 had an affinity with the HOPS complex (*Figure 8A*), as does Stx17 (*Gillingham et al., 2014*). We envision that upon completion of autophagosome biogenesis/maturation, Rab2 and Stx17 are recruited to the outer autophagosomal membrane. Then, the HOPS complex is subsequently recruited to autophagosomes in a Rab2-depedent manner through coincident interactions with both Stx17 and Rab2 (*Figure 9D–E*). At the same time, the HOPS complex binds Rab7 on lysosomes. In turn, the HOPS complex tethers autophagosomes and lysosomes to promote trans-SNARE complex formation between Stx17, SNAP29 and Vamp7/8 and ultimately autophagosome-lysosome fusion (*Figure 9F*).

We found that the Rab2 role in autophagy discovered in fly muscle relates to a broader autophagy requirement in other cell types and across species. The localization of Rab2 on autophagosomes in Drosophila IOMs was conserved for both Rab2A and Rab2B in mouse embryonic fibroblasts (MEFs). As in flies, the Rab2A/2B double knockout led to a delay or block in autophagy clearance as indicated by accumulation of LC3/Atg8. However, the specific Rab2 loss-of-function phenotypes were not identical. While Rab2 was required for autophagosome-lysosome fusion in fly IOMs, the Rab2A/2B double knockouts in MEFs indicated a requirement at a later step in autophagic clearance. Interestingly, this disparity in autophagy phenotypes across species is also seen with Rab7. In flies and yeast, Rab7/Ypt7 is essential for autophagosome-lyosome/vacuole fusion (*Kirisako et al., 1999*; *Hegedűs et al., 2016*), while mammalian Rab7 knockdowns more clearly indicate a required role in autolysosome maturation (*Gutierrez et al., 2004*; *Jäger et al., 2004*). Other examples indicate that the autophagosome-lysosome fusion machinery is not highly evolutionarily conserved. The Stx17-SNAP29-VAMP7/8 trans-SNARE complex is conserved in Drosophila and mammals, but not in yeast (*Shen and Mizushima, 2014*), where no autophagosomal SNARE has been reported so far. Moreover, budding yeast do not encode for Rab2 (*Klöpper et al., 2012*).

Altogether, it is plausible that Rab2 is required for autophagosome-lysosome fusion efficiency, and Rab2-dependency is variable across different tissues or species. Two possible models could explain the different Rab2 autophagy requirements in flies and mouse cells. First, it is suggested that autophagosomes sequentially fuse with endosomes then lysosomes to become amphisomes and autolysosomes, respectively. If either of the steps requires Rab2A/2B, then intermediates with partially degraded contents could accumulate in double knockout MEFs. Alternatively, an autophagosome may normally fuse with multiple lysosomes to ensure full degradation of its contents (*Tsuboyama et al., 2016*). In the absence of Rab2A/2B in MEFs, autophagosomes could still fuse but not with a sufficient number of lysosomes, resulting in an accumulation of partially digested autolysosomes.

Rab2 has been previously associated with transport events at the Golgi apparatus, ER-to-Golgi traffic and secretory granule formation (*Ailion et al., 2014*; *Edwards et al., 2009*; *Saraste, 2016*), as well as in a *C. elegans* endocytic/phagocytic pathway (*Chun et al., 2008*; *Mangahas et al., 2008*). Gillingham *et al.* systematically explored Rab effectors in Drosophila cultured cells, and found that Rab2 interacts with the HOPS complex besides known Golgi-resident effectors (*Gillingham et al., 2014*). The interaction between Rab2 and HOPS complex is also conserved in mammals (*Kajiho et al., 2016*) (*Figure 8A*), and we found the unexpected Rab2 localization to autophagosomes (*Figure 8B–H* and *Figure 8—figure supplement 1*). Thus, it is likely that Rab2 exerts multiple functions through interaction with different effectors at different places. We do not exclude a possible Rab2 function in the endosome-lysosome system that affects autophagic flux, although we did not detect clear lysosomal defects in Rab2A/B knockout MEFs (*Figure 7H–I* and *Figure 7—figure supplement 2F–I*). Several other factors that localize to autophagosomes or late endosomes-lysosomes, including Atg14, PLEKHM1 and EPG5 (*Tian et al., 2010*; *Tabata et al., 2010*; *Carlsson and Simonsen, 2015*), have been shown to control autophagosome maturation. It is plausible that Rab2 contributes to autophagosome maturation through both a direct role in the fusion

mechanism and an indirect role in endo-lysosome maturation, the same as Rab7 and the HOPS complex.

How Rab2 localizes to autophagosomes remains unclear. Localization of Rab2 on autophagosomes in IOMs did not depend on HOPS complex subunits, Vps39 and Vps41, or on Stx17 (*Figure 9B–C*). Further studies will be needed to determine the identities of the Rab2 guanine nucleotide exchange factor (GEF) and GTPase-activating protein (GAP) that regulate Rab2 GTPase activity in autophagosome-lysosome fusion. A conserved TBC domain protein, OATL1/TBC1D25, is a strong candidate for a Rab2 GAP, given OATL1 localization to autophagosomes and involvement in autophagosome-lysosome fusion (*Itoh et al., 2011*). Further, it was reported that OATL1 directly bound to and showed GAP activity for Rab2A (*Itoh et al., 2006*).

Autophagy is critical for the maintenance of myofiber homeostasis in mammalian skeletal muscle. It is known that several myopathies are associated with excess accumulation of autophagic structures in muscle (*Malicdan et al., 2008*). Further, loss of autophagy in mouse skeletal muscle shows anomalies, including abnormal mitochondria, disassembled sarcomeres and disorganized triads (*Masiero et al., 2009*), as also seen in aged muscle (*Demontis et al., 2013*; *Weisleder et al., 2006*). It is established that autophagy is down-regulated during the course of aging (*Rubinsztein et al., 2011*). This evidence points to a possible significance of autophagy in myofiber remodeling and in T-tubule maintenance. Jumpy/MTMR14 PI3-phosphatase and Dynamin-2 (DNM2) GTPase, two causative genes of human centronuclear myopathy, are required for not only T-tubule maintenance but also proper progression of autophagy (*Dowling et al., 2010*; *Durieux et al., 2012*). Based on these reports and our findings, we speculate that their roles in T-tubule maintenance are mediated, at least in part, through autophagy.

Signaling pathways that regulate atrophy and hypertrophy in Drosophila have been identified (*Piccirillo et al., 2014*), however, the mechanisms and direct mediators of muscle remodeling remain largely unknown. IOM remodeling is a good model to study the mechanisms of muscle remodeling, given that the signaling pathways that control muscle remodeling are conserved between Drosophila and mammals (*Piccirillo et al., 2014*). Advantages of the IOM system are not only its genetic tractability, but also its reproducibility and structure. As a relatively giant single cell along the body wall, IOMs enable tracking of a single cell and its subcellular organization during metamorphosis. Our results show that studies in IOMs can provide new insights into the mechanisms of muscle remodeling as well as regulation of fundamental membrane trafficking pathways, such as autophagy and endocytosis.

## Materials and methods

### Reagents and antibodies

The following reagents were used: Alexa Fluor 546 Phalloidin (1.0 U/ml; Invitrogen/Thermo Fisher Scientific, Waltham, MA), LysoTracker Red DND-99 (1:5,000; Thermo Fisher Scientific, Waltham, MA) and Magic Red Cathepsin B Assay (ImmunoChemistry Technologies, Bloomington, MN). The following antibodies were used: mouse anti-fly Dlg1 (1:200; clone 4 F3; Developmental Studies Hybridoma Bank, Iowa City, IA; RRID:AB_528203), rabbit anti-mouse Atg16L1 ([*Itoh et al., 2011*]; RRID:AB_2631282), rabbit anti-mouse Rab2A (*Aizawa and Fukuda, 2015*) (1:1000; RRID:AB_2631288), rabbit anti-mouse Rab2B (*Aizawa and Fukuda, 2015*) (1:1,000; RRID:AB_2631289), rat anti-fly Atg8 (*Takáts et al., 2013*) (1:500; gift of G. Juhasz; RRID:AB_2568972), rabbit anti-Ref(2)P (*Pircs et al., 2012*) (1:500, also called p62; gift of G. Juhasz; RRID:AB_2569199), rabbit anti-Zormin B1 (*Burkart et al., 2007*) (1:500; B. Bullard; RRID:AB_2631283), guinea pig anti-fly Dor (1:500; gift of H. Krämer; RRID:AB_2569525), rabbit anti-human LC3 (1:50 for immuno-EM; 1:1000 for IF; 1:3000 for WB; PM036, MBL, Woburn, MA; RRID:AB_2274121), mouse anti-GFP JL-8 (1:25 for immuno-EM; Clontech, Mountain View, CA; RRID:AB_10013427), mouse anti-T7 (1:3,000; EMD Millipore Novagen, Danvers, MA; RRID:AB_11211744), mouse anti-FLAG (1:3,000; clone M2; Sigma, St. Louis, MO; RRID:AB_439685), mouse anti-Myc (1:1000 for IF and 1:3000 for WB; clone 9E10, Santa Cruz Biotechnology, Dallas, TX; RRID:AB_627268), rat anti-mouse Lamp1 (1:1,000; clone 1D4B, Santa Cruz Biotechnology, Dallas TX; RRID:AB_2134495), sheep anti-EGFR (1:2,000; #20-ES04, Fitzgerald Industries, Acton, MA; RRID:AB_231428) and mouse anti-$\beta$-actin (1:5,000; Applied Biological Materials, Canada; RRID:AB_2631287).

## Drosophila strains

Flies were reared at 25°C, unless stated. For muscle-targeted gene expression, DMef2-GAL4 driver was used. *UAS-LacZ* was used as a control in RNAi experiments. Genotypes used in this study include the following: (1) *y w; P{w[+mC]=GAL4-Mef2.R}3* (Bloomington Drosophila Stock Center, Bloomington, IN; BDSC 27390), (2) *w; P{w[+mC]=UAS-mCD8::GFP.L}LL5, P{UAS-mCD8::GFP.L}2* (BDSC 5137), (3) *w; Rtnl:GFP$^{G00071}$* (FlyTrap), (4) *w; P{w[+mC]=UASp-GFP.Act79B}3–1* (BDSC 9247), (5) *w; UASp-mCherry:GFP:Atg8a,* (from I. Nezis and H. Stenmark), (6) *yw; UAS-GFP:Atg8a* (from T. Neufeld), (7) *w;UAS-GFP:Lamp1* (from H. Krämer), (8) *w; UAS-GFP:Rab7* (from M. Gonzalez-Gaitan), (9) *y w; P{w[+mC]=UAST-YFP.Rab2}l(3)neo38[02]* (BDSC 23246), (10) *w; UAS-IR-Rab2$^{GD11158}$* (Vienna Drosophila Resource Center, Austria; VDRC 34767; Rab2 RNAi), (11) *y v; P{y [+t7.7] v[+t1.8]=TRiP.JF02377}attP2* (Transgenic RNAi Project, TRiP; BDSC 27051; Rab7 RNAi), (12) *y v; P{y[+t7.7] v[+t1.8]=TRiP.JF01937}attP2* (TRiP, BDSC 25896; Stx17 RNAi), (13) *y v; P{y[+t7.7] v [+t1.8]=TRiP.JF01883}attP2* (TRiP, BDSC 25862; SNAP29 RNAi), (14) *w; UAS-IR-Vamp7$^{NIG.1599R-1}$* (NIG-Fly, Japan; Vamp7 RNAi), (15) *y sc v; P{y[+t7.7] v[+t1.8]=TRiP.HMS02438}* (TRiP, BDSC 42605; Vps39 RNAi), (16) *w; UAS-IR-Vps18$^{KK102176}$* (VDRC 107053; Vps18 RNAi), (17) *w; UAS-IR-Vps11$^{KK102566}$* (VDRC 107420; Vps11 RNAi), (18) *w; UAS-IR-Atg1* (from G. C. Chen; Atg1 RNAi) (*Chen et al., 2008*), (19) *w; UAS-IR-Atg18$^{KK100064}$* (VDRC 105366; Atg18 RNAi), (20) *w; UAS-IR-Atg3$^{KK108666}$* (VDRC 101364; Atg3 RNAi). New genotypes generated during this study include the following: (1) *w; UASt-GFP:Stx17$^4$*, (2) *w; UASt-mCherry:Stx17$^{10}$*, (3) *w; UAS-mCD8:GFP; DMef2-GAL4, UAS-Dcr2*, (4) *w; DMef2-GAL4, UAS-Dcr2*, (5) *w; UAS-mCherry:GFP:Atg8a; DMef2-GAL4, UAS-Dcr2*, (6) *w; UAS-Dcr2; DMef2-GAL4, UAS-GFP:Stx17$^4$*, (7) *w; UAS-GFP:Atg8; DMef2-GAL4, UAS-mCh:Stx17$^3$*, (8) *w; DMef2-GAL4, UAS-YFP:Rab2$^{02}$*, (9) *w; UAS-mCherry:Atg8/CyO; DMef2-GAL4, UAS-YFP:Rab2$^{02}$*, (10) *w; DMef2-GAL4, UAS-IR-Rab2$^{GD11158}$*, (11) w; DMef2-GAL4, sqh-EYFP:Mito$^{BDSC7194}$*, (12) *w; DMef2-GAL4, UAS-GFP:Tubby-Cter.*

## DNA engineering

The pMRX-IRES-puro retroviral vector was a kind gift from S. Yamaoka (*Saitoh et al., 2003*). To generate recombinant retroviruses, cDNAs corresponding to GFP:Rab7, GFP:Rab2A, GFP:Rab2B, mStrawberry:Rab2A and mStrawberry:Rab2B were subcloned into the pMRX-IRES-puro vector. pMRX-IRES-Puro-SECFP:Stx17TM, pMRXIP-SECFP:LC3 have been previously described (*Itakura et al., 2012*; *Koyama-Honda et al., 2013*). pEF-FLAG tag expression vectors carrying cDNAs of mouse Rabs were prepared as described previously (*Fukuda, 2003*). Full length mouse Vps39 and Vps41 were PCR-amplified from mouse cDNAs and each cloned into pEF-BOS T7 vector (*Fukuda et al., 1999*) to generate pEF-BOS T7-Vps39 and pEF-BOS T7-Vps41. Full length Drosophila Syntaxin17 was PCR amplified from cDNA, cloned into pENTR/D-TOPO (Life Technologies/ Thermo Fisher Scientific, Waltham, MA) and subcloned by an LR recombination into the Gateway destination vectors to generate pUASt-EGFP:Stx17 and pUASt-mCherry:Stx17. Transgenic flies expressing UASt-GFP:Stx17 or UASt-mCh:Stx17 were generated following standard injection procedures (BestGene, Inc., Chino Hills, CA).

## RNAi screens of IOM remodeling in Drosophila

In primary RNAi screen, RNAi inverted repeat (IR) hairpins (*Supplementary file 1*) were crossed to *w; DMef2-GAL4, UAS-Dcr2* at 25°C. Lethality was checked after two weeks from crossing. The adult progenies were incubated for 1 week at 25°C. Around twenty flies in the vials were tapped to the bottom, and the speed at which they climbed of the vial was checked. To identify factors required for IOM remodeling, *w; UAS-mCD8:GFP; DMef2-GAL4, UAS-Dcr2* flies were crossed with IR lines that induced a defect in eclosion or mobility in the primary screen. Pharate adults were removed from the pupal case and mounted without dissection. IOMs were observed from dorsal side through the cuticle for z-series imaging by confocal microscopy. At least ten IOMs in three animals were checked for each genotype.

## Muscle preparations and immunofluorescence in Drosophila

Muscle preparations in pupal abdomens were performed as previously described (*Ribeiro et al., 2011*) with several modifications. Staged pupae were removed from the pupal case and pinned on a sylgard-covered petri dish in dissection buffer (5 mM HEPES, 128 mM NaCl, 2 mM KCl, 4 mM

MgCl$_2$, 36 mM sucrose, pH 7.2). Abdomens were opened with scissors, pinned flat, and fixed for 20 min. (4% formaldehyde, 50 mM EGTA, PBS). Then, the samples were unpinned and blocked for 30 min (0.3% bovine serum albuminum (BSA), 2% goat serum, 0.6% Triton, PBS), incubated with primary antibody overnight at 4°C, washed (0.1% Triton PBS), then incubated for 2h at room temp with Alexa Fluor-conjugated secondary antibodies (Molecular Probes/Thermo Fisher Scientific, Waltham, MA) and counterstained with phalloidin for F-actin as needed. The stained samples were washed and mounted in SlowFade reagent (Life Technologies/Thermo Fisher Scientific, Waltham, MA).

## Cell culture and recombinant retroviruses

Plat-E cells (RRID:CVCL_B488) were provided by T. Kitamura (*Morita et al., 2000*). Immortalized wildtype mouse embryonic fibroblasts (MEFs) were provided by M. Komatsu (*Ichimura et al., 2008*). Parental, Rab2A-KO, Rab2B-KO or Rab2A/B double KO MEFs were grown in Dulbecco's modified Eagle's medium (DMEM) supplemented with 10% fetal bovine serum, 2 mM L-glutamine, 5 U/ml penicillin, and 50 U/ml streptomycin in a 5% CO$_2$ incubator at 37°C. Recombinant retroviruses were prepared as previously described (*Saitoh et al., 2003*). Stable transformants were selected in growth medium with 1.5 µg/ml puromycin (Invivogen, San Diego, CA). COS-7 cell line was obtained from ATCC (Manassas, VA; RRID:CVCL_0224). All cell cultures were routinely checked and found to be negative for mycoplasma contamination as determined by DNA (DAPI) staining.

## Starvation and immunofluorescence in MEFs

For starvation, MEFs were quickly rinsed with PBS and incubated in EBSS for 1h (immunostaining) or 2h (LC3-II flux assay). For immunofluorescence, cells were cultured on coverslips, fixed with 4% PFA in PBS for 10 min, and permeabilized with 50 µg/ml digitonin in PBS for 5 min. Samples were then blocked with PBS containing 3% BSA for 15 min. Primary antibodies were diluted 1:500 to 1:1000, and Alexa Fluor–conjugated secondary antibodies (Invitrogen/Thermo Fisher Scientific, Waltham, MA) were diluted 1:1000 in PBS containing 3% BSA. Coverslips were incubated with primary antibodies for 1h, washed five times with PBS, and incubated with secondary antibodies for 45 min. Samples were mounted using ProLong Gold Antifade Mountant (Thermo Fisher Scientific, Waltham, MA).

## CRISPR/Cas9 gene knockout

Guide RNAs for Rab2A or Rab2B knockout in MEFs were designed using CRISPR Direct (http://crispr.dbcls.jp/) and a pair of corresponding oligonucleotides was synthesized for each: Rab2A-KO sgRNA sense sequence, 5'-CACCGtacatcatcatcggcgacac-3' and Rab2A-KO sgRNA antisense sequence, 5'-AAACgtgtcgccgatgatgatgtaC-3'; Rab2B-KO sgRNA sense sequence, 5'-CACCGtccttcagtttaccgacaag-3' and Rab2B-KO sgRNA antisense sequence, 5'-AAACcttgtcggtaaactgaaggaC-3'. The CRISPR/Cas9-mediated Rab2A or 2B knockout was performed according to the protocol by (*Ran et al., 2013*). Briefly, the sgRNA expression constructs were prepared by cloning annealed sgRNA into linearized pSpCas9 (BB)−2A-Puro vector (Addgene, Cambridge, MA). The obtained pSpCas9-Rab2A/B KO plasmid was transfected into wild-type MEFs. 24h post-transfection, transformants were selected by 1.0 µg/mL puromycin (Calbiochem, San Diego, CA). After another 48h, the cells were trypsinized and cloned by limited dilution. Clonal lines were isolated and analyzed through immunoblotting with specific anti-Rab2A or anti-Rab2B antibody to detect endogenous Rab2A/B protein level.

## Western blots

Equal amounts of proteins per sample were subjected to SDS-PAGE, and transferred to polyvinylidene difluoride membranes. The membranes were blocked with PBST (PBS and 0.1% Tween 20) containing 1% skim milk and were then incubated overnight at 4°C with primary antibodies 1000–3000x diluted in the blocking solution. Membranes were washed three times with PBST, incubated for 1h at room temperature with 20,000x dilutions of HRP-conjugated secondary antibodies (GE Healthcare Life Sciences, Pittsbugh, PA) in the blocking solution, and washed five times with PBST. Immunoreactive bands were then detected using ECL plus (GE Healthcare Life Sciences, Pittsburgh, PA) and X-ray films.

## Confocal fluorescence microscopy

For imaging of live pupal IOMs, staged pupae were removed from the pupal case, mounted between slide-glass and cover-glass following a protocol (*Zitserman and Roegiers, 2011*), and imaged from the dorsal side. Live or immunostained IOMs were acquired on a Zeiss LSM 700 microscope with a 10x air/0.45 NA Plan Apochromatic objective or 40x oil/1.3 NA Plan Apochromatic objective. Image acquisition software used was Zen (Carl Zeiss Microscopy, Germany). For imaging in MEFs, fixed and immunostained samples were observed on a FV1000D confocal microscope with a 60x oil/1.4 NA Plan Apochromatic objective. Image acquisition software used was Fluoview (Olympus Life Science, Japan). The images were adjusted using the Photoshop CS4 (Adobe, San Jose, CA) and ImageJ software (https://imagej.nih.gov/ij/).

## Mosaic analysis in MEFs

To compare parental (wild-type) and Rab2A/B DKO MEFs side by side, parental MEFs and Rab2A/B_DKO MEFs stably expressing EGFP were mixed and plated on coverslips. The resultant mosaic samples were immunostained for LC3, stained with Lysotracker Red (5000 times diluted) for 5 min or stained with Magic Red (300 times diluted) for 15 min.

## Electron microscopy

For TEM of IOMs, staged pupae were removed from pupal cases, pinned on a sylgard-covered petri dish, dissected in fixative (2% paraformaldehyde, 2.5% glutaraldehyde, 150 mM sodium cacodylate, pH 7.4) and fixed for 1.5h at room temp. For TEM of fat body, staged larvae were treated as published (*Jean et al., 2015*), ripped open in fixative and incubated 25 min at room temp in an eppendorf tube. Fixative was replaced, samples fixed an additional 20 min, and then fat body tissue was dissected from the fixed larval carcass into glass vials. For TEM of MEFs, cells grown in complete medium were fixed in 2% glutaraldehyde (GA) in 0.1 M sodium cacodylate buffer (pH7.4) for overnight at 4°C, gently scraped in the fixative and then pelleted down by centrifugation. Then, samples were post-fixed in 1% osmium tetroxide in 0.1 M cacodylate buffer and stained in 1% uranyl acetate for 1h. Abdomen fillets were embedded in epoxy resin and 70 nm sections were collected on Formvar and carbon-coated copper grids. Images were acquired on a transmission electron microscope (FEI Tecnai Spirit G2 BioTWIN, Hillsboro, OR) and photographed by a bottom mount Eagle 4 K digital camera (Sony, Japan).

## Immunoelectron microscopy

MEFs stably expressing GFP:Rab2A were cultured in EBSS for 90 min and fixed in 4% paraformaldehyde in 0.1 M Sorensen's phosphate buffer pH 7.4 for overnight at 4°C. Fixed cells were washed with 0.15 M glycine/phosphate buffer, embedded in 10% gelatin/phosphate buffer and infused with 2.3 M sucrose/phosphate buffer overnight at 4°C. One $mm^3$ cell bocks were mounted onto specimen holders and snap frozen in liquid nitrogen. Ultracryomicrotomy was carried out at –100°C on a Leica Ultracut UCT with EM FCS cryoattachment (Leica, Bannockburn, IL) using a Diatome diamond knife (Diatome US, Hatfield, PA). Eighty to ninety nm frozen sections were picked up with a 1:1 mixture of 2.3 M sucrose and 2% methyl cellulose (15cp) as described by *Liou et al. (1996)*, and transferred onto Formvar and carbon-coated copper grids. Immunolabeling was performed by a slight modification of the 'Tokuyasu technique' *Tokuyasu, 1980*. Briefly, grids were placed on 2% gelatin at 37°C for 20 min, rinsed with 0.15 M glycine/PBS and the sections were blocked using 1% cold water fishskin gelatin. Ultrathin cryosections were labeled with mouse anti-GFP (1:25) and rabbit anti-LC3 (1:50) in 1% BSA/PBS for 2h at room temperature, followed by 1h incubation with 12 nm gold conjugated goat anti-mouse IgG and IgM (Jackson ImmunoResearch, West Grove, PA) and 18 nm gold conjugated goat anti-rabbit IgG (both 1:20). Sections were contrasted 10 min in 0.4% uranyl acetate and 1.8% methylcellulose on ice then imaged using a JEOL 1200EX II transmission electron microscope (JEOL, Peabody, MA) and photographed using a Gatan digital camera (Gatan, Pleasanton, CA).

## Live imaging

Live-cell fluorescence imaging was performed using DeltaVison Elite (GE Healthcare Life Sciences, Pittsburgh, PA) equipped with a 60x PlanAPO oil-immersion objective lens (Olympus Life

Science, Japan, NA 1.42) and a cooled-CCD camera (Photometrics, Tucson, AZ, CoolSNAP HQ2). Cells stably expressing marker proteins fused with GFP variants were placed on a glass bottom dish (Greiner Bio-One, Monroe, NC, 617870) 2 days before imaging. During live-cell imaging, the dish was mounted in a chamber (TOKAI HIT, Japan, INUB-ONI-F2) to maintain the incubation conditions at 37°C and 5% $CO_2$. Images were acquired at intervals of 1 min. Time series of 16-bit images were converted into 8-bit Tiff images using ImageJ software (Rasband, W.S., ImageJ, U. S. National Institutes of Health, Bethesda, Maryland, USA, http://imagej.nih.gov/ij/, 1997–2015; RRID:SCR_003070) and the LOCI bio-formats ImageJ plugin.

## EGFR degradation assay

MEFs were serum-starved for overnight and incubated in serum-free DMEM containing 100 ng/ml EGF and 25 μg/ml cycloheximide for the indicated times (*Jean et al., 2015*). Cells were lysed and subjected to SDS–PAGE and immunoblotting. Three independent experiments were performed and quantified.

## Statistics

Each experiment was performed at least three times as biological and technical replicates (at least three different cohorts of unique flies or cell culture populations were analyzed in repeat procedures performed on at least three different days). One exception was for TEM analyses, which were performed on two parallel replicates with multiple animals each. All replicate experiments were performed in parallel with wildtype controls. RNAi phenotypes for the 'Rab2 class' were fully penetrant. Data shown is representative and inclusive of all results for procedures with properly performing experimental controls. Pearson correlation measurements for co-localization quantification were performed with ImageJ software (https://imagej.nih.gov/ij/) for *Figure 1D*, *Figure 4B* and *Figure 4D*. For each combination, ten 30 × 30 μm cropped images taken from different animals were used. Quantification of autophagic flux with mCherry:GFP:Atg8a was performed as previously described (*Takáts et al., 2013*). Quantification of the number of autophagosomes, autolysosomes or mitochondria in TEM images was performed as follows: double-membrane bound structures with undigested cytoplasmic contents (autophagosomes), single membrane bound structures with electron dense cytoplasmic contents (autolysosomes) or mitochondria were quantified manually and normalized to the area of section as shown in *Figure 4F*, *Figure 5C*, *Figure 7G* and *Figure 7—figure supplement 1F*. For each genotype, ten images taken from multiple animals were used. GFP:Atg8 puncta in IOMs were quantified manually and then normalized to the IOM area for *Figure 6B and D*. To measure T-tubule area in *Figure 6D*, we manually set a threshold where the dimmest tubules were visible and plasma membrane and other brighter structures were excluded. More than 10 images were used for the quantification. LC3 puncta quantification in *Figure 7C* was performed as previously described (*Itoh et al., 2011*) using ImageJ. Thirty cells were quantified for each condition. Statistical analysis was performed using a two-tailed unpaired $t$ test. Colocalization of puncta was counted manually in *Figure 8F and H* and *Figure 8—figure supplement 1*, only taking into account overlapping structures with a similar shape in relevant fluorescent channels. Ten images were quantified for each combination. p-values<0.05 were considered statistically significant.

## Acknowledgements

We are grateful to B Bullard, G Juhasz, H Krämer, GC Chen, S Yamaoka, T Kitamura, A Miyawaki, H Stenmark, I Nezis, T Neufeld, M Gonzalez-Gaitan, Bloomington Drosophila Stock Center, DGRC, VDRC, FlyTrap, and NIG fly for reagents. We thank T Meerloo, Y Jones, V Taupin and M Farquhar for technical assistance and use of the UCSD CMM Electron Microscope Facility. We are grateful to members of the Kiger lab and Fukuda lab for helpful comments. This work was supported in part by JSPS, the Uehara Memorial Foundation, and the Kanae Foundation postdoctoral fellowships to NF, Grant-in-Aid for Scientific Research on Innovative Areas grant 25111005 to NM, Grant-in-Aid for Scientific Research from the Ministry of Education, Culture, Sports, and Technology grant 16 H01189 to MF, and an American Heart Association grant 15IRG22830029 to AK.

# Additional information

### Competing interests

IK-H: Reviewing editor, *eLife*. The other authors declare that no competing interests exist.

### Funding

| Funder | Grant reference number | Author |
| --- | --- | --- |
| American Heart Association | Innovative Research Grant,15IRG22830029 | Amy A Kiger |
| Japan Society for the Promotion of Science | Postdoctoral Fellowship | Naonobu Fujita |
| Uehara Memorial Foundation | Postdoctoral Fellowship | Naonobu Fujita |
| Kanae Foundation | Postdoctoral Fellowship | Naonobu Fujita |
| Japan Society for the Promotion of Science | Grant-in-Aid for Scientific Research,25111005 | Noboru Mizushima |
| Ministry of Education, Culture, Sports, Science, and Technology | Grant-in-Aid for Scientific Research,16H01189 | Mitsunori Fukuda |

The funders had no role in study design, data collection and interpretation, or the decision to submit the work for publication.

### Author contributions

NF, Designed the research, performed the experiments, analyzed and interpreted data, and wrote the manuscript with input from all authors; WH, T-hL, J-FG, SJ, JN, Performed the experiments and analyzed and interpreted data; YK, IK-H, NM, Analyzed and interpreted data; MF, Generated reagents and analyzed and interpreted data.; AAK, Designed the research, analyzed and interpreted data, and wrote the manuscript with input from all authors.

### Author ORCIDs

Noboru Mizushima, http://orcid.org/0000-0002-6258-6444
Mitsunori Fukuda, http://orcid.org/0000-0002-8620-5853
Amy A Kiger, http://orcid.org/0000-0003-4300-176X

# Additional files

### Supplementary files

• Supplementary file 1. RNAi lines screened with DMef2-GAL4 muscle-targeted expression.

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
