## [Decision Letter]

[Editors’ note: a previous version of this study was rejected after peer review, but the authors submitted for reconsideration. The first decision letter after peer review is shown below.]

Thank you for submitting your work entitled "Genetic screen in *Drosophila* muscle identifies autophagy-mediated T-tubule remodeling and a Rab2 role in autophagy" for consideration by *eLife*. Your article has been favorably evaluated by K VijayRaghavan (Senior Editor) and three reviewers, one of whom is a member of our Board of Reviewing Editors. The reviewers have opted to remain anonymous.

Our decision has been reached after consultation between the reviewers. Based on these discussions and the individual reviews below, we regret to inform you that your work will not be considered further for publication in *eLife*.

The reviewers agreed that the two main findings of this study, namely that autophagy plays a role in regulating T-tubule remodeling and that the small GTPase Rab2 mediates autophagosome maturation, are novel and interesting. However, this study falls short of providing mechanistic information to meet the standards for *eLife*. A substantial amount of work needs be conducted to address the concerns raised by the reviewers, and it is unlikely that this can be done within the two-month limit set by *eLife*. I do encourage you to resubmit your work after you have obtained mechanistic data to support the function of autophagy in T-tubule remodeling or the role of Rab2 in autophagosome maturation.

*Reviewer #1:*

In this manuscript, Fujita et al. examined the dynamics of T-tubule membrane structures in dorsal internal oblique muscles (IOMs). They found that T-tubule membranes undergo disassembly-reassembly and uncovered a role of autophagy in this remodeling process. Autophagy is required for progression through T-tubule disassembly stages. The authors then used IOMs to screen factors required for T-tubule remodeling and revealed that RNAi of Rab2, Rab7, STX17, SNAP29 or VAMP7/8 causes accumulation of small vesicles positive for the T-tubule marker mCD8. They further provided evidence to show that Rab2 is essential for autophagosome maturation. Rab2 localizes to autophagosomes and binds to the HOPS complex. This manuscript includes two relatively separate parts, one on autophagy in T-tubule remodeling and another on the role of Rab2 in autophagosome tethering/fusion. However, no mechanistic insights have been shown in either part. In general, this study fails to meet the stringent requirements for publication in *eLife*.

1) What is the role of autophagy in T-tubule membrane remodeling? It has been shown that disassembled T-tubule membranes are delivered into an endomembrane trafficking pathway. Does autophagy control this process? Are the T-tubule membranes contributing to autophagosomal membranes for degradation of mitochondria during T-tubule remodeling? The detailed function of autophagy in T-tubule remodeling should be examined.

2) The role of Rab2 in the autophagosome tethering/fusion process is not firmly supported by the data. The impaired autophagosome maturation could result from the defective trafficking processes at the ER and Golgi, in which Rab2 function is essential. HOPS is involved in many vesicle trafficking processes. How and when is Rab2 recruited to autophagosomes? Is Rab2 essential for targeting HOPS to autophagosomes?

3) The autophagy defect appears to be distinct in Rab2-depleted flies and Rab2A/B DKO MEFs. In flies, loss of Rab2 function causes accumulation of unfused autophagosomes, while nondegradative autolysosomes accumulate in Rab2A/B DKO MEFs. If Rab2 acts with HOPS for tethering/fusion of autophagosomes, the defect in Rab2A/B DKO cells should be similar to that in HOPS KD cells. Do autophagosomes or autolysosomes accumulate in HOPS KD cells?

4) Localization of Rab2 on autophagosomes should be detected by immunoEM assays.

5) The degradation capacity of lysosomes in Rab2A/B DKO cells could be examined.

6) Figure 7. Tagged proteins are used to demonstrate the interaction of Rab2 with HOPS and colocalization of Rab2 with LC3. Endogenous proteins should be used for these IP and IF experiments.

*Reviewer #2:*

T-tubules are membrane invaginations from the sarcolemma to the cell interior which forms a tortuous network and is essential for muscle excitation-contraction. In *Drosophila*, the T-tubules disassemble and reassemble with internal oblique muscle remodeling during metamorphosis, however, the underlying mechanism remains elusive. In this study, authors performed RNAi screen for Rab GTPases, Arf GTPases, sorting nexins, BAR domain proteins, SNARE proteins, and phosphoinositide regulators that may affect T-tubule membrane remodeling. Rab2, Rab7, Stx17, SNAP29, VAMP7/8 and components of HOPS complex functions were found to be involved in T-tubule remodeling. They further demonstrated that autophagy is essential for the progression of T-tubule disassembly stages and uncovered a conserved role for Rab2 in the autophagic process. Overall, the findings are interesting, however, the precise function of autophagy in T-tubule remodeling remains unclear. My specific comments are as below:

1) It has been shown that autophagic cell death plays a pivotal role in larval tissues degeneration during *Drosophila* metamorphosis. The authors should determine whether cell death also plays a role for IOM and T-tubule remodeling during metamorphosis.

2) Is there any genetic interaction between rab2 and factors known to shape the T-tubule network such as CAV3, DYSF, BIN1/Amph2, MTM1, DNM2 during T-tubule remodeling?

3) Rab GTPases cycle between GTP-bound active and GDP-bound inactive form. Does Rab2 inactive form have the same phenotype as Rab2-RNAi in T-tubule remodeling and autophagy? How about Rab2 constitutive activation?

4) The authors should quantify the results of Figure 4.

5) Figure 6, In addition to MEFs, the authors should also check the role of Rab2 and autophagy in during mouse myoblast differentiation, fusion and myotube formation. This will provide further insights on their findings in *Drosophila*.

*Reviewer #3:*

Two main findings are reported in this manuscript, namely that autophagy mediates developmentally regulated T-tubule remodeling and that the small GTPase Rab2 mediates autophagy. Through an RNAi screen of 300 lines, the authors found that silencing of Rab2, Rab7, Syntaxin17, Syntaxin7/8 or SNAP29 resulted in a similar phenotype with the T-tubule marker mCD8:GFP found on small vesicles in IOMs. The four latter proteins have previously been implicated in autophagy, and the authors found that silencing of Rab2 also inhibits autophagic flux. Autophagy was indeed found to be upregulated during early metamorphosis, which coincided with IOM remodeling, and muscle-targeted silencing of autophagy proteins resulted in disorganized and fragmented T-tubules in IOMs along with accumulation of mitochondria. The authors also found that knockout of Rab2A and Rab2B gives an autophagic phenotype in MEFs, confirming a conserved role for Rab2 in autophagy. Both Rab2A and Rab2B were found to co-IP with components of the mammalian HOPS complex, and co-localization between Rab2 and Atg8/LC3 were found in both IOMs and mammalian cells. The authors conclude that autophagy plays a crucial role in T-tubule remodeling and propose that Rab2 recruits the HOPS complex to autophagosome membranes to mediate autophagosome-lysosome fusion. Both the two main findings in this manuscript are novel and interesting, but some of the data need to be strengthened in order to support the conclusions.

1) The evidence that Rab2 is present on autophagosome membranes (Figure 7) is not very convincing. Even though LC3/Atg8 is an autophagosome marker, this protein is also present on amphisomes and autolysosomes. The supplement to Figure 7 seems to exclude the possibility that Rab2 is present on autolysosomes, but its possible presence on amphisomes has not been addressed. Furthermore, the ER is known to form contacts with autophagosomes, and at the level of the light microscope it cannot be excluded that Rab2 co-localizing with LC3 is present on ER elements rather than on autophagosomes.

2) The authors need to provide better evidence that Rab2 interacts with the HOPS complex to promote autophagosome-lysosome fusion. Is HOPS recruitment to autophagosomes abolished upon Rab2 silencing?

3) Appropriate quantifications are lacking in Figure 4 and Figure 5.

4) In the absence of relevant *Drosophila* mutants it would be useful to demonstrate rescue of key RNAi phenotypes by siRNA-resistant constructs.

---

## [Author Response]

*[…]*

*Reviewer #1:*

[…]

*1) What is the role of autophagy in T-tubule membrane remodeling? It has been shown that disassembled T-tubule membranes are delivered into an endomembrane trafficking pathway. Does autophagy control this process? Are the T-tubule membranes contributing to autophagosomal membranes for degradation of mitochondria during T-tubule remodeling? The detailed function of autophagy in T-tubule remodeling should be examined.*

We performed further experiments to determine the relationship between T-tubule membrane remodeling and autophagy. We demonstrate that autophagy is required for progression of T-tubule membrane disassembly. Collectively, our results show that autophagy acts immediately downstream of the initiation and membrane scission of T-tubule disassembly to play a key role in further subsequent disassembly and mobilization of T-tubule-derived membranes. Moreover, a block in autophagy-mediated T-tubule disassembly blocked further T-tubule membrane remodeling/reassembly.

First, we performed a more focused timecourse over the first day of wildtype myofiber remodeling in order to precisely assess the order of both T-tubule remodeling and autophagy events. In intact pupal abdominal muscles, we used live imaging and quantification of GFP markers for T-tubule/plasma membrane or autophagosomes, including GFP:TubbyC for PI(4,5)P2 detection, mCD8:GFP and GFP:Atg8, respectively. From this, we now show that there is precise coincidence in the onset and peak timing of T-tubule disassembly and autophagy induction by 18h-20h after pupal formation (APF) (new data, Figure 6).

Second, we assessed the timing requirement for autophagy in T-tubule remodeling by monitoring a similar timecourse in autophagy-disrupted myofibers. In all cases, disruption of either autophagy initiation (*Atg1, Atg3* or *Atg18* RNAi) or autophagosome clearance (*Rab2, Rab7* or *Stx17* RNAi) resulted in initial T-tubule membrane phenotypes that appeared at 18h APF and persisted as a block in remodeling at this stage (new data, Figure 6). While initiation and membrane scission of T-tubule disassembly were indicated by proper timing in the onset of membrane reorganization (Figure 6) and subsequent lack of T-tubule membranes continuous with the plasma membrane (now Figure 4 and Figure 5), the disruption of autophagy functions consistently altered the pattern in T-tubule-derived membrane at/after 18h APF. Interestingly, the genetic block of autophagy initiation (*Atg1, Atg3, Atg18* RNAi) resulted in an accumulation of mCD8:GFP-positive stacked membranes that enlarged over time in the absence of autophagy, while block of a step in autophagy clearance (*Rab2, Rab7, Stx17* RNAi) resulted in an accumulation of mCD8:GFP-marked autophagosomes. Altogether, autophagy is required for proper disassembly and subsequent mobilization of T-tubule membranes, potentially as a contributing source to autophagosome membranes.

In addition, we uncovered an autophagy requirement for mitophagy during myofiber remodeling. We have added additional fluorescence microscopy support showing that a block in mitochondrial clearance occurs in all of the above autophagy-disrupted conditions (new data, Figure 5). This also was indicated by TEM results with accumulated mitochondria either in the cytoplasm (*Atg1* or *Atg18* RNAi) or in nondegraded autophagosomes (*Rab2* or *Rab7* RNAi), respectively (Figure 4, Figure 5). This suggests a specific regulated requirement for mitophagy during pupal muscle remodeling stage, in contrast to previous reports indicating a noticeable lack of mitochondria in autophagosomes upon autophagy induction in larval muscle (Zirin et al.2013). We do not know at this time any significance of mitophagy to T-tubule remodeling. However, we did observe during IOM remodeling the prevalence of both mCD8:GFP-marked autophagosomal membranes and mitochondria as autophagosomal cargo, raising the possible significance of T-tubule membrane disassembly to the process of mitophagy. Testing the relationship between T-tubule and autophagosomal membranes will require future work, including the identification of a means to disrupt T-tubule disassembly independent of autophagy.

To our knowledge, no previous accounts exist on the identity, timing or molecular-genetic nature of a wildtype regulated T-tubule membrane remodeling program. We previously speculated that regulated T-tubule remodeling could occur in IOMs, given our demonstrated remodeling of integrin adhesion complexes in wildtype IOMs and the pupal-specific integrin and T-tubule phenotypes observed with *mtm* RNAi (Ribeiro et al.2011, PLoS Genetics).

*2) The role of Rab2 in the autophagosome tethering/fusion process is not firmly supported by the data. The impaired autophagosome maturation could result from the defective trafficking processes at the ER and Golgi, in which Rab2 function is essential. HOPS is involved in many vesicle trafficking processes. How and when is Rab2 recruited to autophagosomes? Is Rab2 essential for targeting HOPS to autophagosomes?*

We have greatly strengthened support for a Rab2 recruitment and function on mature autophagosomes with the addition of multiple new data. We clearly show in both fly and mouse that Rab2 is recruited to and has required functions at mature autophagosomes. Our new data also shows that Rab2 is recruited to LC3-positive autophagosomes prior to Stx17 recruitment and is required for HOPS (Dor/Vps18) recruitment.

We previously showed by fluorescence microscopy imaging in MEFs that Rab2A/Rab2B colocalized with the majority of LC3-marked autophagosomes, but did not colocalize with Atg16L1 pre-autophagosomes nor Lamp1-postive endolysosomes (data now in Figure 8). Similarly, fly Rab2 colocalized with Atg8 (now Figure 8) and localized on the accumulated autophagosomes in *Stx17* or *Rab7* RNAi conditions (now in Figure 9). Now, we have added new data that further supports Rab2 recruitment to autophagosomes. We show that Rab2A and Rab7 in the same cells exhibit preferential localizations to LC3-marked autophagosomes or Lamp1-positive endolysosomes, respectively (new data, Figure 8—figure supplement 1). Using immuno-EM, we identified GFP:Rab2A and endogenous LC3 colocalized along same membranes of autophagic structures with the typical appearance of autophagosomes carrying undigested cargo (new data, Figure 8—figure supplement 1). Finally, using live cell time-lapse imaging in MEFs, we found that LC3 preceded Rab2A recruitment to the same autophagosomes by several minutes; in turn, Rab2A preceded Stx17 recruitment to the same autophagosomes by several minutes (new data, Figure 9 and Figure 9—figure supplement 1).

To address the requirements for and significance of Rab2 localization at autophagosomes, we performed new molecular epistasis experiments. We now show that Rab2 recruitment to Atg8-positive autophagosomes does not require *Stx17, Rab7* or *Vps39* (HOPS) functions in myofibers (new data, Figure 9). Importantly, we found the converse that *Rab2* RNAi diminished the recruitment of HOPS complex component, Dor/Vps18, to Atg8-marked autophagosomes (new data, Figure 9). Together with the conserved Rab2 protein-protein interactions with Vps39 and Vps41 (now Figure 8) and conserved requirement for late stages in autophagy (now Figure 4, Figure 7), our data strongly supports an autophagosomal role for Rab2.

As this reviewer noted, HOPS is involved in multiple vesicle trafficking processes. Our MEF double knockout results together with protein localization and interaction data suggest that Rab2A/Rab2B may play an essential role in a HOPS-dependent amphisome-lysosome fusion, as well as a role in autophagosome-lysosome fusion (now Figure 7). However, Rab2A/Rab2B double knockout MEFs did not impair general lysosomal function (now Figure 7), as further supported now by new data demonstrating normal EGF-induced EGFR degradation response in double knockout cells (new data, Figure 7—figure supplement 2).

Although our genetic screens were limited in scope, the results did not point to an obvious or similar requirement for ER-Golgi trafficking in T-tubule remodeling ([Supplementary-material SD18-data]). Additionally, we tested 2 of 11 known Rab2 binding proteins, and neither ICA-69 and p115 knockdown phenocopied the Rab2 or Atg1 class of RNAi phenotypes.

While it is challenging to formerly exclude the possibility of additional Rab2 functions elsewhere in the cell that may also contribute to autophagosome maturation, we have shown by multiple methods in two different species a conserved recruitment of and function for Rab2 at mature autophagosomes.

*3) The autophagy defect appears to be distinct in Rab2-depleted flies and Rab2A/B DKO MEFs. In flies, loss of Rab2 function causes accumulation of unfused autophagosomes, while nondegradative autolysosomes accumulate in Rab2A/B DKO MEFs. If Rab2 acts with HOPS for tethering/fusion of autophagosomes, the defect in Rab2A/B DKO cells should be similar to that in HOPS KD cells. Do autophagosomes or autolysosomes accumulate in HOPS KD cells?*

We have not generated MEF knockout cells for HOPS components, so we cannot at this time answer the question of a shared HOPS requirement.

We have discussed what we believe are possible reasonable explanations for phenotypic differences between Rab2 RNAi in fly muscle and Rab2A/2B knockout MEFs (Discussion, seventh paragraph). In addition, we now show that the accumulation of Atg8-autophagosomes and defect in autophagic clearance seen with *Rab2* knockdown in *Drosophila* larval fat body corresponded with an accumulation of undigested autolysosomes detected by TEM (new data, Figure 7—figure supplement 1), indicating parallels between Rab2 knockdown phenotypes in fly fat body and in knockout MEFs. Interestingly, new findings have observed that an individual autophagosome can fuse with multiple lysosomes (Tsuboyama et al., 2016, Science). It is possible that Rab2A/2B could play a role in the efficiency of autophagosome-lysosome fusion. In the absence of Rab2A/2B function, autophagosomes may still fuse but with an insufficient number of lysosomes, resulting in accumulation of partially digested autolysosomes.

*4) Localization of Rab2 on autophagosomes should be detected by immunoEM assays.*

As mentioned above, we now include immuno-EM detection of GFP:Rab2A with endogenous LC3 along the same membranes of autophagic structures with typical appearance of autophagosomes (new data, Figure 8—figure supplement 1).

*5) The degradation capacity of lysosomes in Rab2A/B DKO cells could be examined.*

We now show that Rab2A/Rab2B double knockout MEFs did not impair general lysosomal function, as supported by normal EGF-induced EGFR degradation (new data, Figure 7—figure supplement 2).

*6) Figure 7. Tagged proteins are used to demonstrate the interaction of Rab2 with HOPS and colocalization of Rab2 with LC3. Endogenous proteins should be used for these IP and IF experiments.*

We agree. Unfortunately, antibody reagents for endogenous Rab2 or Rab2A/2B are not available.

*Reviewer #2:*

[…]

*1) It has been shown that autophagic cell death plays a pivotal role in larval tissues degeneration during Drosophila metamorphosis. The authors should determine whether cell death also plays a role for IOM and T-tubule remodeling during metamorphosis.*

In our previous published studies, we demonstrated that cell death is not occurring over the course of wildtype IOM remodeling in pupal stages (Ribeiro et al. 2011, PLoS Genetics). Using mutants that specifically disrupted IOM attachment and number during pupal remodeling, we also showed that viable, functional IOMs are required after the remodeling period to assist with adult eclosion (emergence) from the pupal case. We showed that regulated IOM cell death, however, does occur 1 day after adult eclosion, as previously noted. We have now added reference to this background information and discussion to the text (Introduction, Results and Discussion).

*2) Is there any genetic interaction between rab2 and factors known to shape the T-tubule network such as CAV3, DYSF, BIN1/Amph2, MTM1, DNM2 during T-tubule remodeling?*

We are not aware of genetic interactions between Rab2 and factors known to shape the T-tubule network. This is a very interesting line of inquiry that is now generally feasible to pursue in fly IOMs (at least for the conserved genes of those listed above). However, this will entail a significant amount of research that would over-extend our manuscript (already at nine figures plus supplemental figures), and that we strongly feel is more suitable for exploration as a follow-up manuscript.

*3) Rab GTPases cycle between GTP-bound active and GDP-bound inactive form. Does Rab2 inactive form have the same phenotype as Rab2-RNAi in T-tubule remodeling and autophagy? How about Rab2 constitutive activation?*

Yes, a dominant negative Rab2A-DN construct induced an elevated number of LC3 puncta when overexpressed in COS7 cells (not shown). In flies, expression of a YFP:Rab2-DN construct resulted in a weak IOM phenotype, likely due to the apparently low expression levels.

*4) The authors should quantify the results of Figure 4.*

We have now included quantification of colocalization between GFP and mCherry signal from GFP:mCherry:Atg8 (new data, Figure 4) and of GFP:Atg8 and mCherry:Stx17 (in subsection “Rab2 or Rab7 knockdown blocks autophagosome-lysosome fusion with 16 autophagy-dependent myofiber remodeling”, first paragraph and Figure 4 legend) by Pearson correlation analysis.

*5) Figure 6, In addition to MEFs, the authors should also check the role of Rab2 and autophagy in during mouse myoblast differentiation, fusion and myotube formation. This will provide further insights on their findings in Drosophila.*

It will be interesting to test the autophagy requirements for T-tubule remodeling in vertebrate myofibers. Mouse myoblast cultures may not be the most appropriate system to explore this question, as it is our understanding that C2C12-derived myotubes fail to fully differentiate a well-formed T-tubule network. We believe that this research direction would be best explored with in vivoanimal models and is an extension well beyond the scope of this manuscript.

*Reviewer #3:*

*[…]*

*1) The evidence that Rab2 is present on autophagosome membranes (Figure 7) is not very convincing. Even though LC3/Atg8 is an autophagosome marker, this protein is also present on amphisomes and autolysosomes. The supplement to Figure 7 seems to exclude the possibility that Rab2 is present on autolysosomes, but its possible presence on amphisomes has not been addressed. Furthermore, the ER is known to form contacts with autophagosomes, and at the level of the light microscope it cannot be excluded that Rab2 co-localizing with LC3 is present on ER elements rather than on autophagosomes.*

We have now greatly strengthened data supporting Rab2 localization on mature autophagosomes, as also presented above in response to reviewer #1. We show that Rab2A and Rab7 in the same cells exhibit preferential localizations to LC3-marked autophagosomes or Lamp1-positive endolysosomes, respectively (new data, Figure 8—figure supplement 1). Using immuno-EM, we identified GFP:Rab2A and endogenous LC3 colocalized along the same membranes of autophagic structures with the typical appearance of autophagosomes carrying undigested cargo (new data, Figure 8—figure supplement 1). Finally, using live cell time-lapse imaging in MEFs, we found that LC3 preceded Rab2A recruitment to the same autophagosome ring by several minutes; in turn, Rab2A preceded Stx17 recruitment to the same autophagosome ring by several minutes (new data, Figure 9 and Figure 9—figure supplement 1). Altogether, these results support Rab2 localization at autophagosome membranes.

We agree that there may be additional Rab2 localizations in the cell that contribute to its autophagy functions. While we found little colocalization between Rab2 and Rab7 or Lamp1 (Figure 8—figure supplement 1), it is possible that there are requirements at these less pronounced sites of localization. We discuss this possibility in the text (Discussion, ninth paragraph).

*2) The authors need to provide better evidence that Rab2 interacts with the HOPS complex to promote autophagosome-lysosome fusion. Is HOPS recruitment to autophagosomes abolished upon Rab2 silencing?*

Yes, we now have tested this and found that Rab2 is required for colocalization of the HOPS component, Dor/Vps18, at Atg8-marked autophagosomes (new data, Figure 9). Additionally, Rab2 recruitment to autophagosomes occurs prior to Stx17 (new data, Figure 9) and does not require *Stx17, Rab7* or *Vps39* (HOPS) functions (new data, Figure 9). Together, this suggests that Rab2 is needed to recruit HOPS complex for autophagosome tethering/fusion.

*3) Appropriate quantifications are lacking in Figure 4 and Figure 5.*

We now include quantification of the number of autophagosomes or the number of mitochondria per IOM area in TEM micrographs (new data, Figure 4 and Figure 5).

*4) In the absence of relevant Drosophila mutants it would be useful to demonstrate rescue of key RNAi phenotypes by siRNA-resistant constructs.*

Yes, this is an important experiment. We now include demonstration of *Rab2* RNAi phenotypic rescue with co-expression of wildtype Rab2 cDNA (new data, Figure 3—figure supplement 1).